# Data leakage inflates prediction performance in connectome-based machine learning models

Matthew Rosenblatt [1] ✉, Link Tejavibulya [2], Rongtao Jiang [3], Stephanie Noble [3,4,5] & Dustin Scheinost [1,2,3,6,7]

Predictive modeling is a central technique in neuroimaging to identify brain-behavior relationships and test their generalizability to unseen data. However, data leakage undermines the validity of predictive models by breaching the separation between training and test data. Leakage is always an incorrect practice but still pervasive in machine learning. Understanding its effects on neuroimaging predictive models can inform how leakage affects existing literature. Here, we investigate the effects of five forms of leakage–involving feature selection, covariate correction, and dependence between subjects–on functional and structural connectome-based machine learning models across four datasets and three phenotypes. Leakage via feature selection and repeated subjects drastically inflates prediction performance, whereas other forms of leakage have minor effects. Furthermore, small datasets exacerbate the effects of leakage. Overall, our results illustrate the variable effects of leakage and underscore the importance of avoiding data leakage to improve the validity and reproducibility of predictive modeling.

Understanding individual differences in brain-behavior relationships is a central goal of neuroscience. As part of this goal, machine learning approaches using neuroimaging data, such as functional connectivity, have grown increasingly popular in predicting numerous phenotypes[1], including cognitive performance[2-6], age[7-10], and several clinically-relevant outcomes[11-13]. Compared to classic statistical inference, prediction offers advantages in replicability and generalizability, as it evaluates models on participants unseen during model training[14,15]. Essentially, the data are split into training and test subsets, such as through k-fold cross-validation or simple train/test splits, so that the model is strictly evaluated on unseen data. Unfortunately, the boundaries between training and test data can be inadvertently violated by data leakage. Data leakage is when information about the test data is introduced into the model during training[16], nullifying the benefits of separating training and test data.

A recent meta-review of machine learning highlighted the prevalence of leakage across 17 fields[17]. Three hundred twenty-nine papers were identified as containing leakage. This meta-review described eight types of leakage: not having a separate test set, preprocessing on the training and test sets, feature selection jointly on the training and test sets, duplicate data points, illegitimate features, temporal leakage, non-independence between the training and test sets, and sampling bias. Data leakage often led to inflated model performance and consequently decreased reproducibility[17]. In another review specific to predictive neuroimaging, ten of the 57 studies may have leaked information by performing dimensionality reduction across the whole dataset prior to train/test splitting[18]. Since leakage may dramatically change the reported results, it contributes to the ongoing reproducibility crisis in neuroimaging[17,19–21]. Despite the prevalence and concern of leakage, the severity of performance

[1]Department of Biomedical Engineering, Yale University, New Haven, CT, USA. [2]Interdepartmental Neuroscience Program, Yale University, New Haven, CT, USA. [3]Department of Radiology & Biomedical Imaging, Yale School of Medicine, New Haven, CT, USA. [4]Department of Bioengineering, Northeastern University, Boston, MA, USA. [5]Department of Psychology, Northeastern University, Boston, MA, USA. [6]Child Study Center, Yale School of Medicine, New Haven, CT, USA. [7]Department of Statistics & Data Science, Yale University, New Haven, CT, USA. ✉e-mail: matthew.rosenblatt@yale.edu

inflation due to leakage in neuroimaging predictive models remains unknown.

In this work, we evaluate the effects of leakage on functional connectome-based predictive models in four large datasets for the prediction of three phenotypes. Specifically, in over 400 pipelines, we test feature leakage, covariate-based leakage, and subject leakage. These leakage types span five of the leakage types described by Kapoor and Narayanan[17] (Supplementary Table 1). We first show the effects of leakage on prediction performance by comparing two performance metrics in various leaky and non-leaky pipelines. Then, we evaluate the effects of leakage on model interpretation by comparing model coefficients. Furthermore, we resample the datasets at four different sample sizes to illustrate how small sample sizes may be most susceptible to leakage. Finally, we extend our analysis to structural connectomes in one public dataset. Overall, our results elucidate the consequences, or in some cases lack thereof, of numerous possible forms of leakage in neuroimaging datasets.

## Results

Resting-state fMRI data were obtained in each of our four datasets: the Adolescent Brain Cognitive Development (ABCD) Study[22] ($N = 7822–7969$), the Healthy Brain Network (HBN) Dataset[23] ($N = 1024–1201$), the Human Connectome Project Development (HCPD) Dataset[24] ($N = 424–605$), and the Philadelphia Neurodevelopmental Cohort (PNC) Dataset[25,26] ($N = 1119–1126$). Throughout this work, we predicted age, attention problems, and matrix reasoning using ridge regression[27] with 5-fold cross-validation, 5% feature selection, and a grid search for the L2 regularization parameter. Specific measures for each dataset are described in the methods

section, but these three broad phenotypes were selected because they are available in all the datasets in this study. In addition, these three phenotypes are appropriate for benchmarking leakage because they span a wide range of effect sizes, with strong prediction performance for age, moderate performance for matrix reasoning, and poor performance for attention problems.

We first evaluated the effects of leakage on prediction in HCPD (Sections "Performance of non-leaky pipelines"–"Subject-level leakage"), and then, we showed effects of leakage in the other three datasets (ABCD, HBN, PNC) (Section "Evaluation of performance in additional datasets"). Moreover, we compared model coefficients (Section "Comparing coefficients in leaky and non-leaky pipelines"), varied the sample size (Section "Effect of sample size"), and performed sensitivity analyses (Section "Sensitivity analyses"). The types of leakage used in this study are summarized in Fig. 1 and further detailed in the "Methods" section.

### Performance of non-leaky pipelines

We evaluated four non-leaky pipelines and found that different analysis choices led to different prediction performance (Fig. 2), as evaluated by Pearson's correlation $r$ and cross-validation $R^2$, also called $q^2$ [28]. Our gold standard model included covariate regression, site correction[29–31], and feature selection within the cross-validation scheme, which was split accounting for family structure. It exhibited no prediction performance for attention problems (median $r = 0.01$, $q^2 = -0.13$), strong performance for age ($r = 0.80$, $q^2 = 0.63$), and moderate performance for matrix reasoning ($r = 0.30$, $q^2 = 0.08$). Notably, $q^2$ may be negative when the model prediction gives a higher mean-squared error than predicting the mean, as was the case for attention problems.

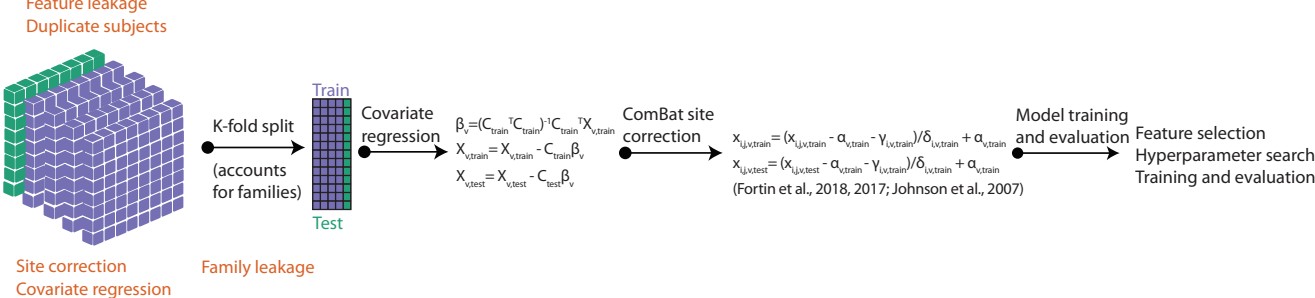

**Fig. 1 | Summary of the prediction pipelines used in this study.** The various forms of leakage that may occur are shown in orange. Feature leakage, leaky site correction, leaky covariate regression, and subject leakage may occur prior to splitting the data into training and test sets. Family leakage may occur during the splitting of data.

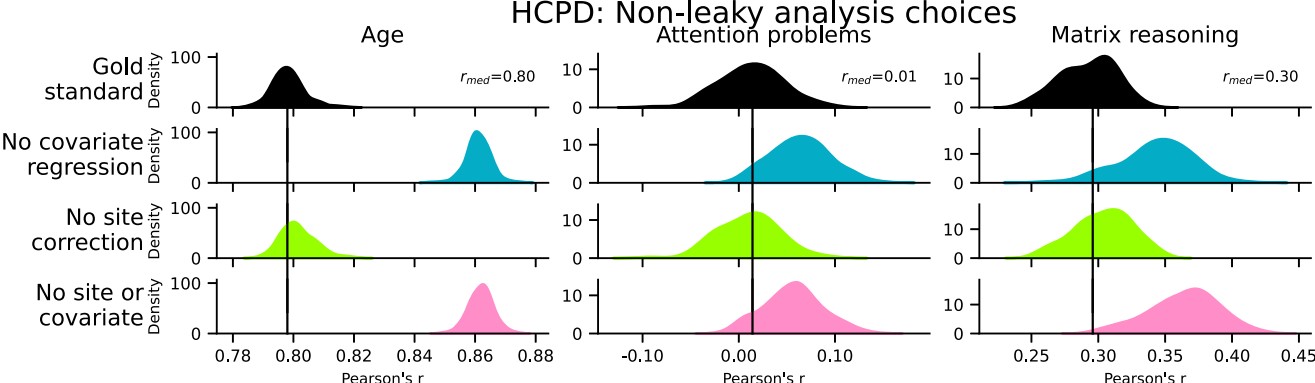

**Fig. 2 | Prediction performance in HCPD of the non-leaky pipelines, including the gold standard, omitting covariate regression, omitting site correction, and omitting both covariate regression and site correction.** Rows represent different non-leaky analysis choices, and columns show different phenotypes. The black bar represents the median performance of the gold standard models across random iterations, and the exact value of the bar is shown as the median $r$, $r_{med}$. The histograms show prediction performance across 100 iterations of 5-fold cross-validation. See also Supplementary Fig. 1. HCPD Human Connectome Project Development.

Performance when excluding site correction was nearly identical to the gold standard model ($|\Delta r| < 0.01$, $\Delta q^2 < 0.01$). However, not regressing out covariates inflated $r$ but had varying effects on $q^2$ for all three phenotypes, including attention problems ($\Delta r = 0.05$, $\Delta q^2 = -0.08$), age ($\Delta r = 0.06$, $\Delta q^2 = 0.11$), and matrix reasoning ($\Delta r = 0.05$, $\Delta q^2 = 0.01$). Though not the main focus of this paper, these results highlight how prediction performance may change with different analysis choices, especially regarding whether or not covariates are regressed from the data.

### Feature leakage

Features should be selected in the training data and then applied to the test data. Feature leakage occurs when selecting features in the combined training and test data. Feature leakage inflated prediction performance for each phenotype (Fig. 3). The inflation was small for age ($\Delta r = 0.03$, $\Delta q^2 = 0.05$), larger for matrix reasoning ($\Delta r = 0.17$, $\Delta q^2 = 0.13$), and largest for attention problems ($\Delta r = 0.47$, $\Delta q^2 = 0.35$). Notably, age had a strong baseline performance and was minimally affected by feature leakage, whereas attention problems had the worst baseline performance and was most affected by feature leakage. Furthermore, the attention problems prediction went from chance-level ($r = 0.01$, $q^2 = -0.13$) to moderate ($r = 0.48$, $q^2 = 0.22$), which highlights the potential for feature leakage to hinder reproducibility efforts.

### Covariate-related leakage

Covariate-related forms of leakage in this study included correcting for site differences and performing covariate regression in the combined training and test data (i.e., outside of the cross-validation folds) (Fig. 4). Leaky site correction had little effect on performance ($\Delta r = -0.01$-$0$, $\Delta q^2 = -0.01$-$0.01$). Unlike the other forms of leakage in this study, leaky covariate regression decreased performance for attention problems ($\Delta r = -0.06$, $\Delta q^2 = -0.17$), age ($\Delta r = -0.02$, $\Delta q^2 = -0.03$), and matrix reasoning ($\Delta r = -0.09$, $\Delta q^2 = -0.08$). These results illustrate that

leakage can not only hamper reproducibility by false inflation of performance, but also by underestimating the true effect sizes.

### Subject-level leakage

Since families are often oversampled in neuroimaging datasets, leakage via family structure may affect predictive models. Given the heritability of brain structure and function[32–34], leakage may occur if one family member is in the training set and another in the test set. Family leakage did not affect prediction performance of age or matrix reasoning ($\Delta r = 0.00$, $\Delta q^2 = 0.00$) but did slightly increase prediction performance of attention problems ($\Delta r = 0.02$, $\Delta q^2 = 0.00$; Fig. 5).

Furthermore, subject-level leakage may occur when repeated measurements data (e.g., multiple tasks) are incorrectly treated as separate participants or when data are accidentally duplicated. Here, we considered the latter case, where a certain percentage of the subjects in the dataset were repeated (called subject leakage), at three different levels (5%, 10%, 20%; Fig. 5). In all cases, subject leakage inflated prediction performance, with 20% subject leakage having the greatest impact on attention problems ($\Delta r = 0.28$, $\Delta q^2 = 0.19$), age ($\Delta r = 0.04$, $\Delta q^2 = 0.07$), and matrix reasoning ($\Delta r = 0.14$, $\Delta q^2 = 0.11$). Similar to the trend seen in feature leakage, the effects of subject leakage were more dramatic for models with weaker baseline performance. Overall, these results suggest that family leakage may have little effect in certain instances, but potential leakage via repeated measurements (i.e., subject leakage) can largely inflate performance.

### Additional family leakage analyses in ABCD

Since the two datasets in this study with family information contained mostly participants without any other family members in the dataset (HCPD: 471/605, ABCD: 5868/7969 participants did not have family members), we performed several additional experiments to determine the effects of family leakage with a larger proportion of families.

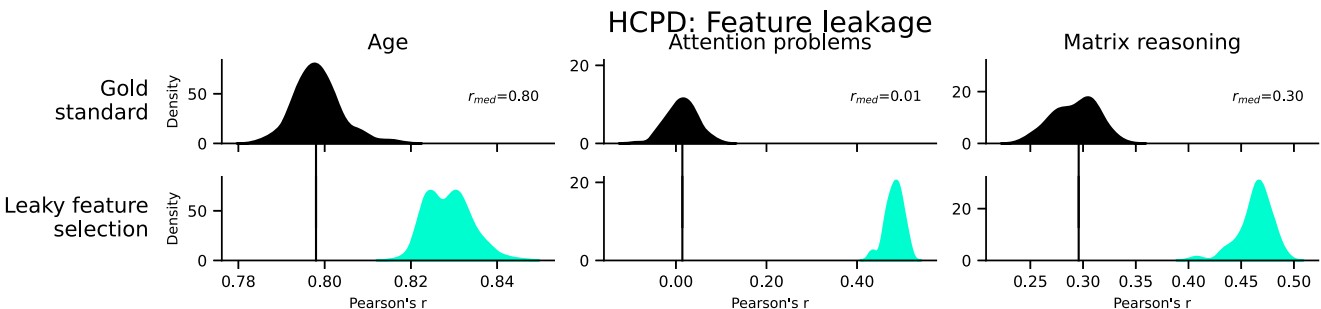

**Fig. 3 | Prediction performance in HCPD of leaky feature selection, compared to the gold standard.** Rows represent different leakage types, and columns show different phenotypes. The black bar represents the median performance of the gold standard models across random iterations, and the histograms show prediction performance across 100 iterations of 5-fold cross-validation. See also Supplementary Fig. 1. HCPD Human Connectome Project Development.

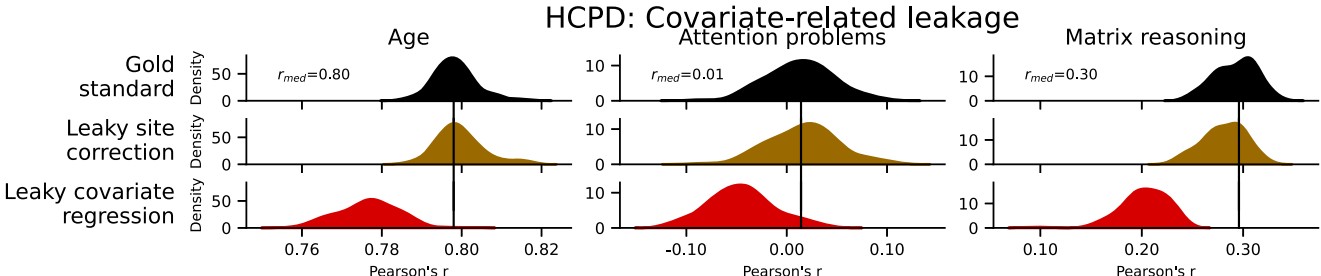

**Fig. 4 | Prediction performance in HCPD of covariate-related forms of leakage, including leaky site correction, and leaky covariate regression.** Rows represent different leakage types, and columns show different phenotypes. The black bar represents the median performance of the gold standard models across random iterations, and the histograms show prediction performance across 100 iterations of 5-fold cross-validation. See also Supplementary Fig. 1. HCPD Human Connectome Project Development.

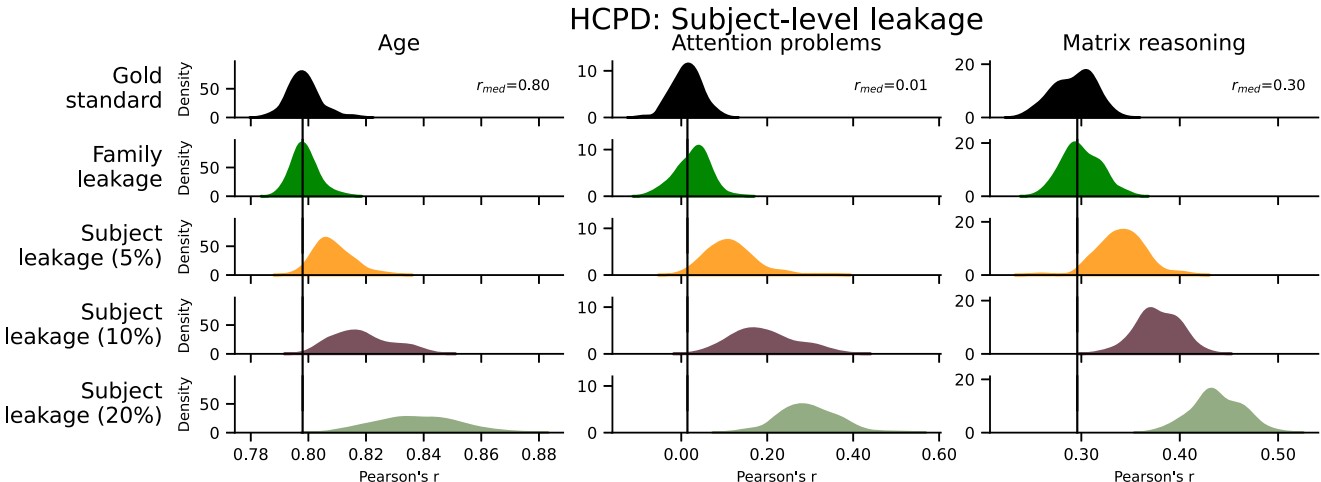

**Fig. 5 | Prediction performance in HCPD of subject-level forms of leakage.** These included family leakage and subject leakage at three different levels. Rows represent different leakage types, and columns show different phenotypes. The black bar represents the median performance of the gold standard models across random iterations, and the histograms show prediction performance across 100 iterations of 5-fold cross-validation. See also Supplementary Fig. 1. HCPD Human Connectome Project Development.

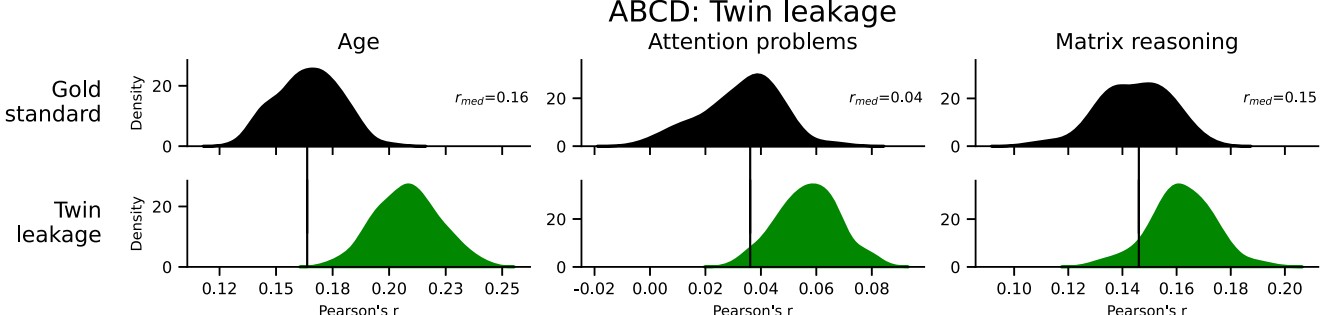

**Fig. 6 | Prediction performance in ABCD comparing the gold standard to twin/family leakage.** The black bar represents the median performance of the gold standard models across random iterations, and the histograms show prediction performance across 100 iterations of 20-fold cross-validation. See also Supplementary Figs. 2–4. ABCD Adolescent Brain Cognitive Development.

We used ABCD instead of HCPD for these experiments because ABCD has more families with multiple members in the dataset.

First, ABCD was restricted to only twins ($N = 563$ pairs of twins, 1126 participants total), after which we performed 100 iterations of 5-fold cross-validation for all three phenotypes and model types. In one case, family structure was accounted for in the cross-validation splits. In another, the family structure was ignored, constituting leakage. Leakage in the twin dataset exhibited minor to moderate increases in prediction performance (Fig. 6), unlike when using the entire dataset. The inflation was $\Delta r = 0.04$ for age and $\Delta r = 0.02$ matrix reasoning and attention problems.

We included several additional phenotypes and models to compare how leakage may affect twin studies (Supplementary Fig. 2), which showed similar results. The phenotype similarity between the twin pairs did not have a strong relationship with changes in performance due to leakage (Supplementary Fig. 3). Furthermore, based on a simulation study, leakage effects increased with the percentage of participants belonging to a family with more than one individual (Supplementary Fig. 4).

### Evaluation of performance in additional datasets
Compared to HCPD, we found similar trends in each of the 11 pipelines for ABCD, HBN, and PNC (Supplementary Figs. 5 and 6). While excluding site correction had little to no effect in HCPD or HBN, there was a small effect in ABCD ($\Delta r = 0.01$–$0.02$, $\Delta q^2 = 0.00$–$0.01$). Furthermore, not performing covariate regression generally inflated performance

relative to the baseline for attention problems ($\Delta r = 0.02$–$0.05$, $\Delta q^2 = -0.08$–$0.04$), age ($\Delta r = -0.01$–$0.06$, $\Delta q^2 = 0.00$–$0.11$), and matrix reasoning ($\Delta r = 0.01$–$0.05$, $\Delta q^2 = -0.02$–$0.01$).

Across all datasets and phenotypes, leaky feature selection and subject leakage (20%) led to the greatest performance inflation. Feature leakage had varying effects based on the dataset and phenotype ($\Delta r = 0.03$–$0.52$, $\Delta q^2 = 0.01$–$0.47$). The dataset with the largest sample size (ABCD), was least affected by leaky feature selection, and weaker baseline models were more affected by feature leakage. Subject leakage (20%) also inflated performance across all datasets and phenotypes ($\Delta r = 0.06$–$0.29$, $\Delta q^2 = 0.03$–$0.24$). Corroborating the results in HCPD, leaky covariate regression was the only form of leakage that consistently deflated performance ($\Delta r = -0.09$–$0.00$, $\Delta q^2 = -0.17$–$0.00$). Family leakage ($\Delta r = 0.00$–$0.02$, $\Delta q^2 = 0.00$) and leaky site correction ($\Delta r = -0.01$–$0.00$, $\Delta q^2 = -0.01$–$0.01$) had little to no effect.

The change in performance relative to the gold standard for each pipeline across all four datasets and three phenotypes is summarized in Fig. 7. Overall, only leaky feature selection and subject leakage inflated prediction performance in this study.

### Comparing coefficients in leaky and non-leaky pipelines
Determining whether the performance of leaky and non-leaky pipelines is similar only tells part of the story, as two models could have similar prediction performance but learn entirely different brain-behavior relationships. As such, establishing how model coefficients may change for various forms of leakage is an equally important facet of understanding

a.

b.

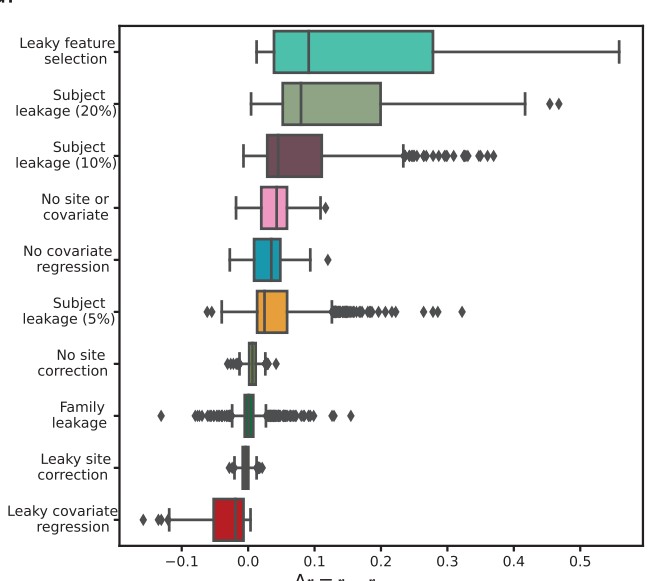

**Fig. 7 | Evaluation of performance differences between all pipelines and the gold standard pipeline across all datasets and phenotypes for Pearson's $r$ and $q^2$.** The plots are ranked from the most inflated performance (top) to the most deflated performance (bottom) by two different performance metrics, **a)** Pearson's $r$ and **b)** $q^2$. Boxplot elements were defined as follows: the center line is the median across all datasets, phenotypes, and iterations (100 per dataset/phenotype combination); box limits are the upper and lower quartiles; whiskers are 1.5x the interquartile range; points are outliers. See also Supplementary Figs. 5 and 6.

the possible effects of leakage. We first averaged the coefficients over the five folds of cross-validation and calculated the correlation between those coefficients and the coefficients of the gold standard model (Fig. 8). Excluding site correction (median $r_{coef} = 0.75$–$0.99$) led to minor coefficient changes. Meanwhile, excluding covariate regression (median $r_{coef} = 0.31$–$0.84$) or excluding both covariate regression and site correction (median $r_{coef} = 0.32$–$0.81$) resulted in moderate coefficient changes. Amongst the forms of leakage, leaky feature selection was, unsurprisingly, the most dissimilar to the gold standard coefficients (median $r_{coef} = 0.39$–$0.72$). Other forms of leakage that notably affected the coefficients included family leakage (median $r_{coef} = 0.79$–$0.94$) and 20% subject leakage (median $r_{coef} = 0.74$–$0.93$). Otherwise, the coefficients between the leaky and gold standard pipelines were very similar. We also compared the coefficients for each pair of 11 analysis pipelines (Supplementary Fig. 7). Interestingly, although the coefficients for excluding covariate regression or performing leaky feature selection were relatively dissimilar to the gold standard coefficients (median $r_{coef} = 0.31$–$0.84$), these coefficients were relatively similar to each other (median $r_{coef} = 0.68$–$0.92$). This result could be explained by covariates contributing to brain-behavior associations in the entire dataset.

In addition to correlating the coefficients at the edge level, we also considered the similarity of feature selection across 10 canonical networks[35] (Supplementary Fig. 8). We calculated the number of edges selected as features in each of the 55 subnetworks, which are defined as a connection between a specific pair of the 10 canonical networks. We then adjusted for subnetwork size and compared the rank correlation across different leakage types. Similar to the previous analysis, not performing covariate regression changed the distribution of features across subnetworks (median $r_{spearman,network} = 0.28$–$0.88$). Amongst the forms of leakage, leaky feature selection showed the greatest network differences compared to the gold standard model (median $r_{spearman,network} = 0.25$–$0.85$), while the other forms of leakage showed smaller differences (median $r_{spearman,network} = 0.75$–$1.00$).

### Effect of sample size
All previously presented results investigated the four datasets at their full sample sizes (ABCD: $N = 7822$–$7969$, HBN: $N = 1024$–$1201$, HCPD:

$N = 424$–$605$, PNC: $N = 1104$–$1126$). Yet, although these can lead to less reproducible results[21], smaller sample sizes are common in neuroimaging studies. As such, consideration of how leakage may affect reported prediction performance at various sample sizes is crucial. For leaky feature selection, leaky site correction, leaky covariate regression, family leakage, and 20% subject leakage, we computed $\Delta r = r_{leaky}$-$r_{gold}$, where $r_{leaky}$ is the performance of the leaky pipeline and $r_{gold}$ is the performance of the gold standard non-leaky pipeline for a single seed of 5-fold cross-validation. For each combination of leakage type, sample size ($N = 100, 200, 300, 400$), and dataset, $\Delta r$ was evaluated for 10 different resamples for 10 iterations of 5-fold cross-validation each (over 20,000 evaluations of 5-fold cross-validation in total; Fig. 9). In general, the variability of $\Delta r$ was much greater for the smallest sample size ($N = 100$) compared to the largest sample size ($N = 400$). For instance, for matrix reasoning prediction in ABCD, $\Delta r$ for family leakage ranged from $-0.34$ to $0.25$ for $N = 100$ and from $-0.12$ to $0.13$ for $N = 400$. Another example is site correction in ABCD matrix reasoning prediction, where $\Delta r$ ranged from $-0.13$ to $0.06$ for $N = 100$ and from $-0.11$ to $0.03$ for $N = 400$. While not every dataset and phenotype prediction had large variability in performance for leaky pipelines at small sample sizes (e.g., HBN age prediction), the overall trends suggest that leakage may be more unpredictable and thus dangerous in small samples compared to large samples.

However, when taking the median performance across multiple k-fold splits for a given subsample, the effects of most leakage types, except feature and subject leakage, decreased (Supplementary Fig. 9). In general, the best practice is to perform at least 100 iterations of k-fold splits, but due to the many analyses and pipelines in this study, we only performed 10 iterations. For instance, for the ABCD prediction of matrix reasoning, taking the median across 10 iterations resulted in a slightly smaller range of $\Delta r$ values for all forms of leakage ($N = 400$), including feature leakage ($\Delta r_{multiple} = 0.17$–$0.67$ for multiple iterations vs. $\Delta r_{single} = 0.10$–$0.71$ for a single iteration), leaky site correction ($\Delta r_{multiple} = -0.11$–$0.03$ vs. $\Delta r_{single} = -0.06$–$0.01$), leaky covariate regression ($\Delta r_{multiple} = -0.08$ to $-0.01$ vs. $\Delta r_{single} = -0.10$–$0.01$), family leakage ($\Delta r_{multiple} = -0.02$–$0.04$ vs. $\Delta r_{single} = -0.12$–$0.13$), and 20% subject leakage ($\Delta r_{multiple} = 0.21$–$0.33$ vs. $\Delta r_{single} = 0.17$–$0.43$). Overall,

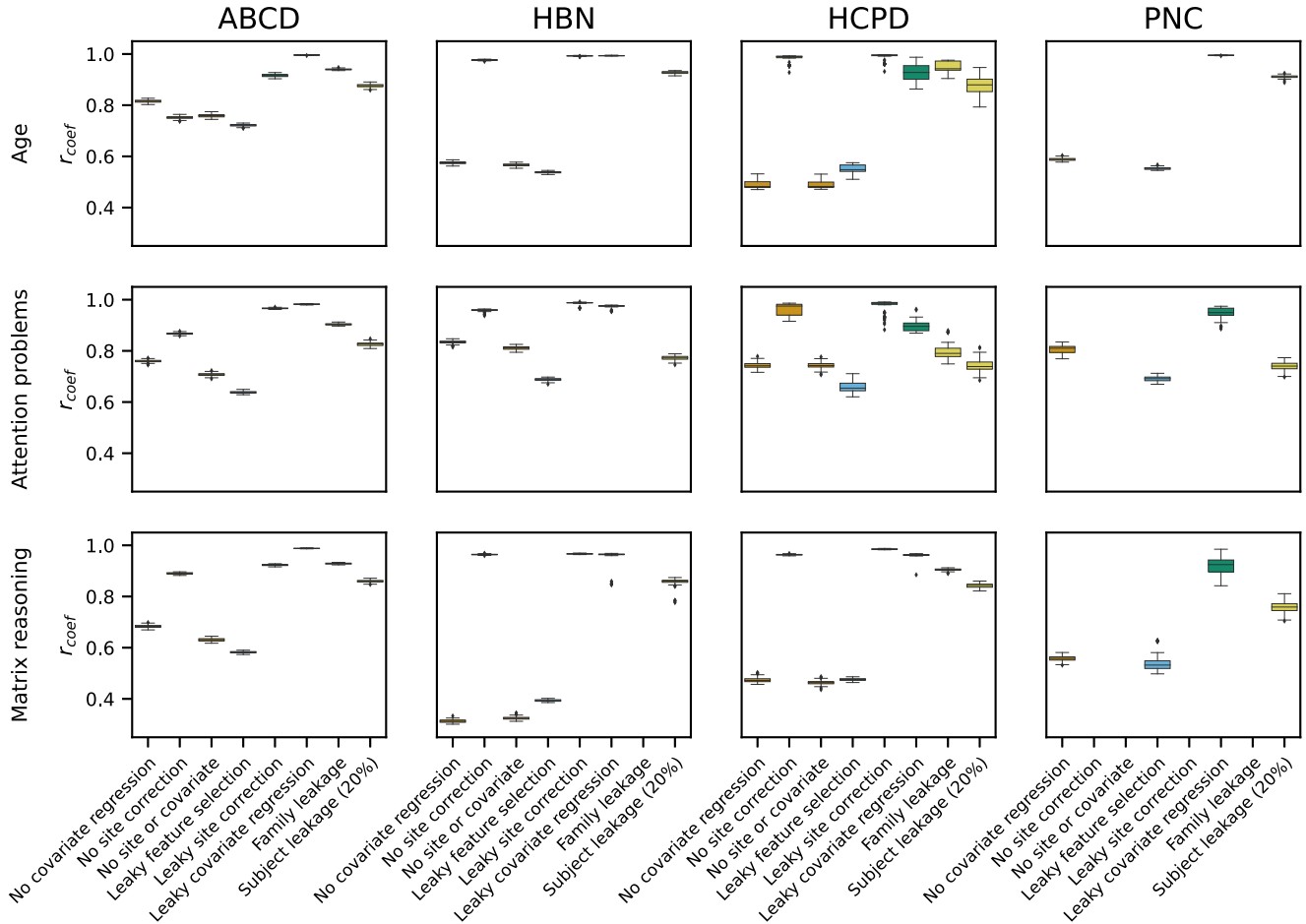

**Fig. 8 | Similarity of coefficients between the gold standard and various forms of leakage.** The boxes are colored by the leakage family: orange (non-leaky analysis choices), blue (feature leakage), green (covariate-related leakage), yellow (subject-level leakage). Boxplot elements were defined as follows: the center line is the median across 100 random iterations; box limits are the upper and lower quartiles; whiskers are 1.5x the interquartile range; points are outliers. Certain values, such as leaky site correction in PNC, are omitted because the relevant fields (e.g., site) do not exist. See also Supplementary Figs. 7 and 8. ABCD Adolescent Brain Cognitive Development, HBN Healthy Brain Network, HCPD Human Connectome Project Development, PNC Philadelphia Neurodevelopmental Cohort.

performing multiple iterations of k-fold cross-validation reduced, but did not eliminate, the effects of leakage. Leakage still led to large changes in performance in certain cases, particularly at small sample sizes.

## Sensitivity analyses

Two main sensitivity analyses were performed to support the robustness of our findings. First, we analyzed the effects of leakage in two other models (SVR, CPM). Second, we performed similar analyses using structural connectomes to demonstrate the effects of leakage beyond functional connectivity.

We repeated the analyses for support vector regression (SVR) (Supplementary Figs. 10 and 12) and connectome-based predictive modeling (CPM)[2] (Supplementary Figs. 11 and 13) and found similar trends in the effects of leakage. CPM generally had a slightly lower gold standard performance (age: median $r = 0.16$–$0.61$, $q^2 = 0.02$–$0.37$; attention problems: $r = -0.04$–$0.11$, $q^2 = -0.25$–$0.00$; matrix reasoning: $r = 0.18$–$0.27$, $q^2 = 0.02$–$0.05$) than ridge regression (age: $r = 0.25$–$0.80$, $q^2 = 0.06$–$0.63$; attention problems: $r = -0.01$–$0.13$, $q^2 = -0.21$–$0.00$; matrix reasoning: $r = 0.25$–$0.30$, $q^2 = 0.06$–$0.08$) and SVR (age: $r = 0.24$–$0.80$, $q^2 = 0.04$–$0.64$; attention problems: $r = 0.00$–$0.12$, $q^2 = -0.15$ to -0.09; matrix reasoning: $r = 0.25$–$0.34$, $q^2 = 0.05$–$0.10$). Notably, CPM was less affected by leaky feature selection ($\Delta r = -0.04$–$0.39$, $\Delta q^2 = 0.00$–$0.38$) compared to ridge regression ($\Delta r = 0.02$–$0.52$, $\Delta q^2 = 0.02$–$0.47$) and SVR ($\Delta r = 0.02$–$0.48$,

$\Delta q^2 = -0.01$–$0.38$). In addition, subject leakage had the largest effect on SVR ($\Delta r = 0.06$–$0.45$, $\Delta q^2 = 0.10$–$0.31$), followed by ridge regression ($\Delta r = 0.04$–$0.29$, $\Delta q^2 = 0.03$–$0.24$) and CPM ($\Delta r = 0.00$–$0.17$, $\Delta q^2 = 0.00$–$0.16$). Regardless of differences in the size of the effects across models, the trends were generally similar.

In addition, we extended our leakage analyses from functional to structural connectomes with 635 participants from the HCPD dataset. Gold standard predictions of matrix reasoning, attention problems, and age exhibited low to moderate performance in the HCPD structural connectome data (Fig. 10 and Supplementary Fig. 11) (matrix reasoning: median $r = 0.34$, $q^2 = 0.12$; attention problems: $r = 0.11$, $q^2 = -0.07$; age: $r = 0.73$, $q^2 = 0.53$). The forms of leakage that most inflated the performance were feature leakage ($\Delta r = 0.07$–$0.57$, $\Delta q^2 = 0.12$–$0.52$) and subject leakage ($\Delta r = 0.05$–$0.27$, $\Delta q^2 = 0.06$–$0.20$). Compared to its effect on functional connectivity data, leaky covariate regression in this particular instance showed milder reductions in performance ($\Delta r = -0.04$–$0.00$, $\Delta q^2 = -0.04$–$0.00$). Despite minor differences, these results in structural connectivity data follow similar trends as the functional connectivity data.

## Discussion

In this work, we demonstrated the effects of five possible forms of leakage on connectome-based predictive models in the ABCD, HBN, HCPD, and PNC datasets. In some cases, leakage led to severe inflation of prediction (e.g., leaky feature selection). In others, there was little to

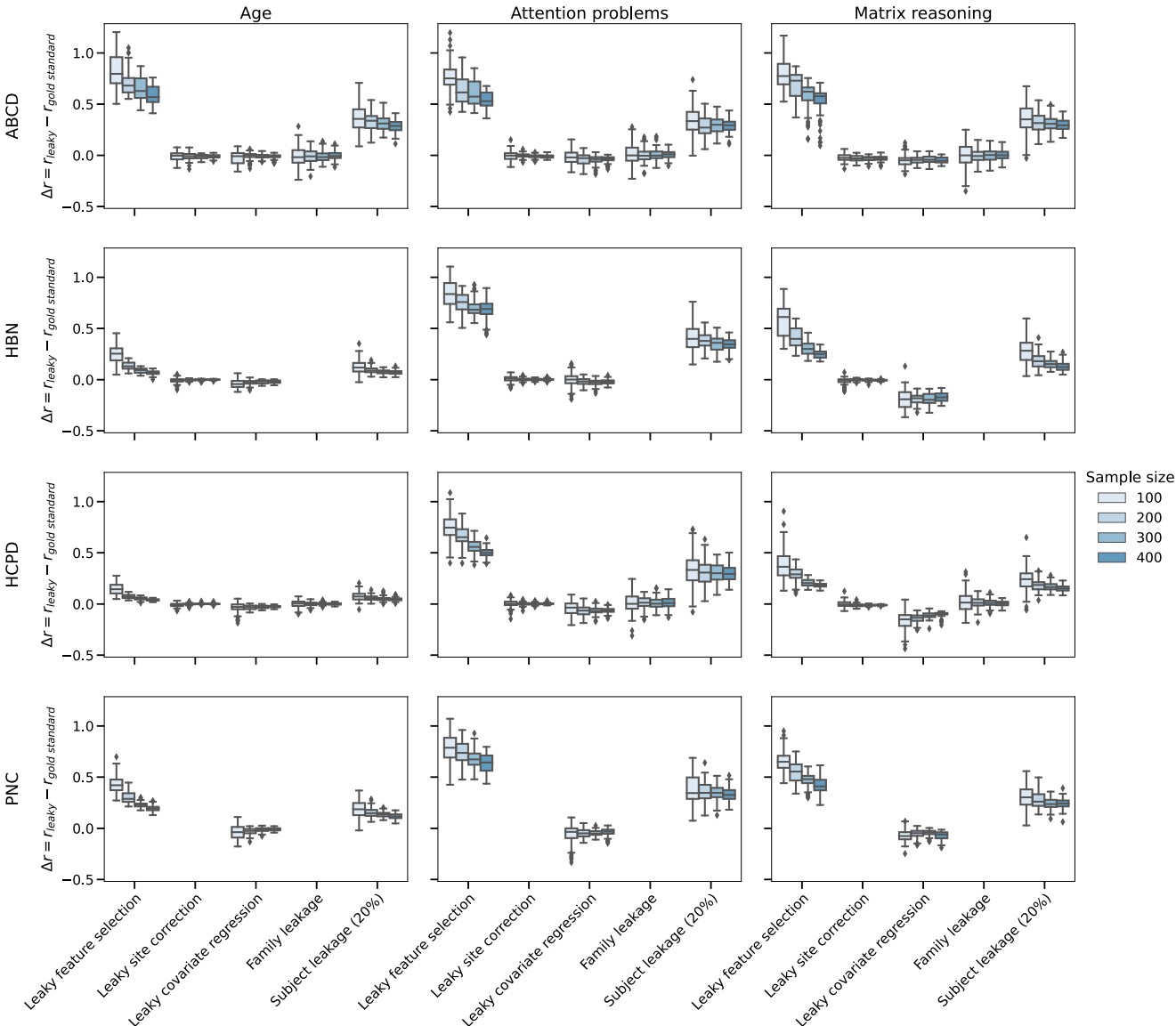

**Fig. 9 | Difference between leaky and gold standard performance for various types of leakage and four sample sizes ($N$ = 100, 200, 300, 400).** Rows represent the dataset, and columns show the phenotype. For each leakage type ($x$-axis), there are four results ($N$ = 100, 200, 300, 400). For each sample size, we repeated 10 random seeds of resampling for 10 iterations of 5-fold cross-validation. Boxplot elements were defined as follows: the center line is the median across all subsampling seeds and cross-validation iterations; box limits are the upper and lower quartiles; whiskers are 1.5x the interquartile range; points are outliers. See also Supplementary Fig. 9. ABCD Adolescent Brain Cognitive Development, HBN Healthy Brain Network, HCPD Human Connectome Project Development, PNC Philadelphia Neurodevelopmental Cohort.

no difference (e.g., leaky site correction). The overall effects of leaky pipelines showed similar trends across the different phenotypes, models, and connectomes investigated in this work. Furthermore, subsampling to smaller sizes—typical in the neuroimaging literature— led to an increased effect of leakage. Leakage is never the correct practice, but quantifying its effects in neuroimaging is still important to understand exactly how much leakage may impede reproducibility in neuroimaging. Given the variable effects of leakage found in this work, the strict splitting of testing and training samples is particularly important in neuroimaging to accurately estimate the performance of a predictive model.

Feature leakage is widely accepted as a bad practice, and as expected, it severely inflated prediction performance. Although feature leakage is likely rare in the literature, it can dramatically increase model performance and thus hinder reproducibility. For instance, a recent work[36] demonstrated that a high-profile article predicting suicidal ideation in youth had no predictive power after removing feature

selection leakage. The original paper, which has now been retracted, received 254 citations since its publication in 2017 according to Google Scholar. As such, it is essential to re-emphasize the importance of avoiding feature leakage. Though avoiding feature leakage may appear obvious, it can occur in more subtle ways. For example, one may investigate which networks significantly differ between two groups in the whole dataset and then create a predictive model using those networks. Notably, the effects of feature leakage were smaller in ABCD due to its large sample size. In other words, when using thousands of samples, the selected features are likely robust across different folds of training data. This result is consistent with recent findings regarding association studies[21]. In general, feature leakage can be reduced by sharing code on public repositories. Although it requires additional work, we strongly urge authors to share their analysis code in all cases and preprocessed data when appropriate. Then, the community can quickly and easily reproduce results and look for potential leakage in the code.

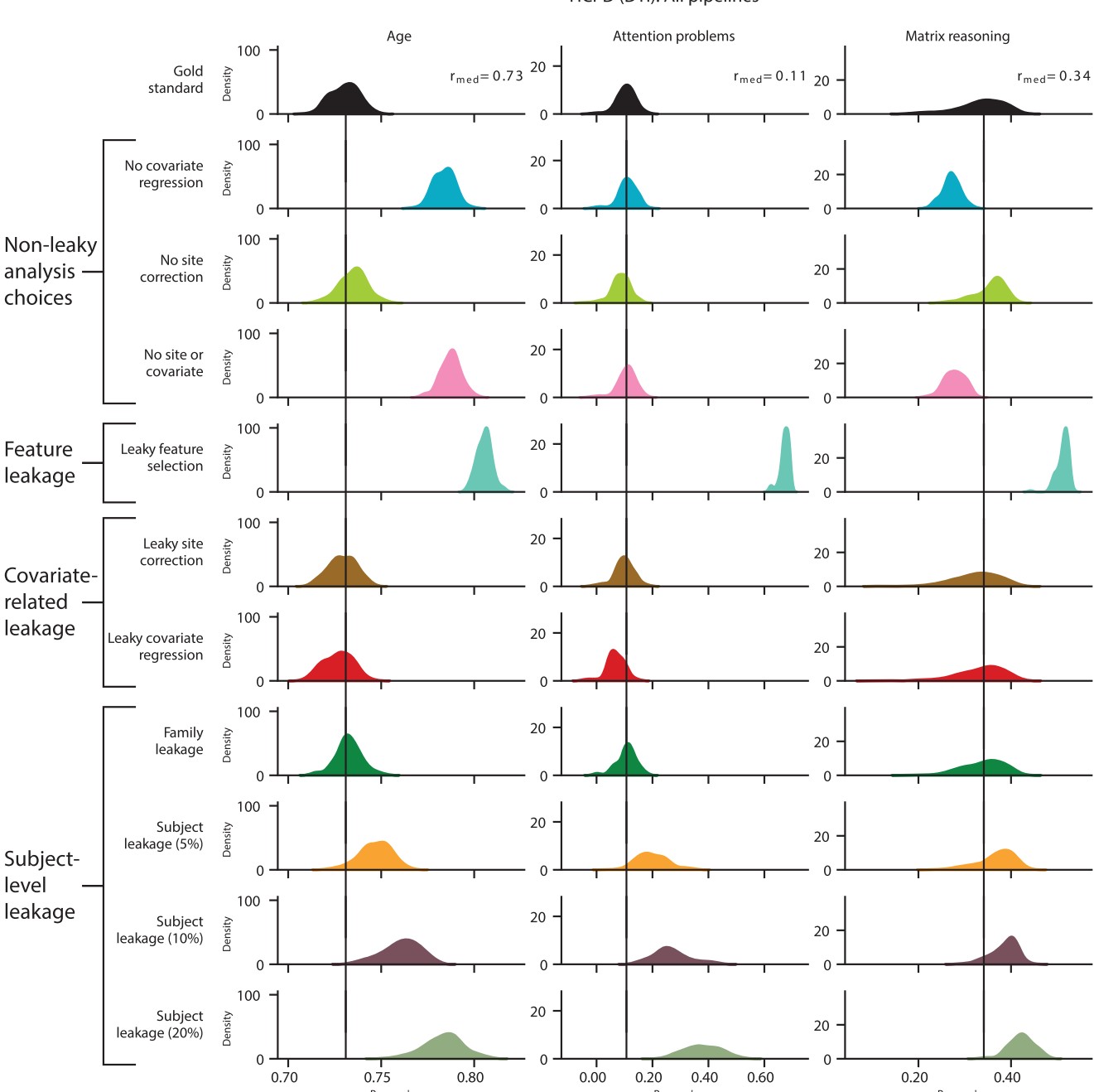

**Fig. 10 | Evaluation of leakage types for matrix reasoning, attention problems, and age prediction in structural connectomes using *r*.** The rows show different leakage types, and the columns show different phenotypes. The black bar represents the median performance of the gold standard models across random iterations, and the histograms show prediction performance across 100 iterations of 5-fold cross-validation. See also Supplementary Fig. 14. HCPD Human Connectome Project Development, DTI diffusion tensor imaging.

Similarly, subject leakage led to inflated effects. It is more likely in datasets with multiple runs of fMRI, time points, or tasks. For example, a high-profile preprint used deep learning to predict pneumonia from chest X-rays, and the authors did not account for patients having multiple scans, causing leakage across the training and test sets. Fortunately, this leakage was identified by a community member and quickly corrected in a subsequent version of the preprint[37], which illustrates the importance of writing detailed methods and sharing code. Extra care should be taken for prediction when using multiple scans per individual, such as when collecting multiple task scans or longitudinal data.

We typically associate leakage with inflated prediction performance. However, leaky covariate regression deflated prediction performance. Our results corroborate previous works, suggesting that covariate regression must be performed within the cross-validation loop to avoid false deflation of effect sizes[38–40]. Interestingly, performing covariate regression itself may lead to leakage[41] and is another consideration when deciding whether and how to implement covariate regression.

Aside from feature and subject leakage, no other leakage significantly increased prediction performance when using large sample sizes. Notably, leaky site correction did not affect prediction performance. Family leakage had little to no effect when using the entire dataset due to the small percentage of participants belonging to families with more than one individual. Still, a twin subset and various

simulations demonstrated that the effects of family leakage are more pronounced when datasets have a larger proportion of multi-member families. Large, public datasets, like ABCD and the broader HCP lifespan datasets, are increasingly multi-site with complex nested dependency between participants (i.e., twins)[42]. These factors facilitate larger sample sizes for better statistical power and more representative samples, which can minimize model bias[1,43–45]. However, accounting for these factors can rapidly increase the complexity of a prediction pipeline. Thus, these results are reassuring for the broader field. Overall, they likely mean that results with these forms of leakage remain valid, at least in these datasets and phenotypes. Although, there is no guarantee that any of these forms of leakage will not inflate performance. Thus, avoiding data leakage is still necessary to ensure valid results.

Among the phenotypes in this work, age, which had the best prediction performance, was the least affected by leakage. This result suggests that leakage may moreso affect phenotypes with a weaker brain-behavior association. When there is a strong brain-behavior relationship, a model may capture the relevant pattern regardless of leakage. However, when there is a weak brain-behavior relationship, the model may predominantly rely on patterns arising from leakage, thus possibly leading to the larger effects of leakage in behaviors with weak effect sizes. In other words, when the effects are very weak (e.g., attention problems in this study), leakage appears to overtake the true effect. Because effect sizes in brain-behavior association studies are often weak, attention to leakage is particularly important. Yet, it is important to note that the effects of leakage were tested in three phenotypes in this study, and do not comprehensively test across all effect sizes.

Crucially, leakage exhibited more variable effects in smaller datasets. As such, accounting for leakage is even more crucial with small samples. All researchers should avoid leakage, but those using small clinical samples or patient groups should be particularly careful. Taking the median performance of the models over multiple iterations of k-fold cross-validation (i.e., different random seeds) mitigated the inflation. This example underscores the benefits of performing many (≥100) iterations of k-fold cross-validation[28,46]. Though k-fold cross-validation is the most common form of evaluation in neuroimaging, train/test splits are not uncommon. Given that train/test splits are often only performed for one random seed, leakage with small sample sizes may be a greater issue when using train/test splits.

Along with its effects on performance, we found that leakage also affected model interpretation, and therefore neurobiological interpretation. Coefficients for feature leakage were unsurprisingly dissimilar to the gold standard because the leaky feature selection relies on one feature subset, whereas the gold standard pipeline selects a different subset of features for each fold of cross-validation. Otherwise, the most notable differences in coefficients arose from omitting covariate regression. This result highlights that, in addition to avoiding leakage, researchers should consider how various analysis choices may affect results[47,48].

The results presented in this work focus on neuroimaging, specifically functional and structural connectivity prediction studies. However, the lessons from this work may be valuable to any field using scientific machine learning. Since we expect that leakage will be prevalent across many fields for the foreseeable future, quantifying the effects of leakage may provide valuable field-specific insight into the validity of published results. Thus, we encourage researchers in other fields to use their domain-specific knowledge to identify possible forms of leakage and subsequently evaluate their effects. Although quantifying the effects is essential due to the pervasiveness of leakage, researchers should pay careful attention to avoiding leakage.

Numerous strategies can help prevent leakage in neuroimaging and other machine-learning applications. These strategies include carefully developing and sharing code, alternative validation strategies, model information sheets[17], skepticism about one's own results, and cross-disciplinary collaborations. Writing and maintaining code should incorporate several facets to reduce the likelihood of leakage, including establishing an analysis plan prior to writing code, using well-maintained packages, and sharing code. One's analysis plan should be set ahead of time, either informally or, if appropriate, formally through pre-registration. As one tries more pipelines, especially if searching for a significant result (i.e., p-hacking), leakage is more likely to occur. A predefined plan could minimize the likelihood of leakage by detailing how features will be selected, which models will be trained, and how possible covariates and nested structures will be handled. Another suggestion for reducing the likelihood of leakage is using well-maintained packages. For example, Scikit-learn has a k-fold cross-validation package[27] that has been thoroughly tested, whereas developing k-fold cross-validation code from scratch may lead to accidental leakage. Among many other benefits, sharing code, particularly well-documented code, could decrease the effects of leakage by allowing external reviewers to investigate published pipelines for leakage. Relatedly, although not always possible, distributing preprocessed data can make the reproduction of results much easier and less time-consuming for reviewers or those who want to verify the validity of a predictive model.

Moreover, most neuroimaging papers are evaluated with train/test splits or k-fold cross-validation. However, alternative validation strategies, such as a lockbox[49] or external validation, may reduce the likelihood of leakage. Both these strategies help to maintain a clearer separation between training and test data, where a lock box entails leaving out a subset of the data until a final evaluation[49] and external validation consists of applying a model to a different dataset. Another strategy to decrease the prevalence of leakage is using model information sheets, such as the one proposed by Kapoor and Narayanan[17]. Model information sheets allow for the authors, reviewers, and public to reflect upon the work and identify possible leakage. However, it may be difficult to verify the accuracy of model information sheets when data cannot be shared[17]. This limitation is especially true for neuroimaging datasets, which often require applications to access the data. As a result, we also recommend healthy skepticism of one's results. For instance, if a machine learning pipeline leads to a surprising result, the code should be scrutinized by asking a collaborator to view one's code or repeat the analyses on synthetic data. Finally, collaborations across disciplines to incorporate domain and machine learning experts will help prevent leakage[17]. Domain experts can bring knowledge of the nuances of datasets (e.g., the prevalence of family structures in neuroimaging datasets). In contrast, machine learning experts can help domain experts train models to avoid leakage.

While this study investigated several datasets, modalities, phenotypes, and models, several limitations still exist. In many instances in this work, leakage had little to no effect on the prediction results, yet this finding does not mean that leakage is acceptable in any case. Another limitation is that this study cannot possibly cover all forms of leakage for all datasets and phenotypes. Other possible forms of leakage, like leakage via hyperparameter selection[49], were not considered in this study, as detailed in the "Methods" section. Furthermore, we studied child, adolescent, and young adult cohorts in well-harmonized datasets in this work, but differences in populations and dataset quality could alter the effects of leakage. For example, we showed that family leakage had greater effects in twin studies. As another example, in the case of site correction, if a patient group was scanned at Site A and the healthy control group at Site B, then site leakage would likely have a large impact. Alternative methods may be more appropriate for accounting for possible covariates or site differences in a prediction setting, such as comparing neuroimaging data models to models built only using covariates or leave-one-site-out prediction[50]. Nevertheless, we still included covariate regression and

site correction in our analysis because they are common in the field and may still be well-suited for using prediction to explain the generalizability of brain-behavior relationships. Also, differences in scan lengths between datasets may drive differences in performance across datasets. However, it should not affect the main conclusions of this paper regarding leakage in machine learning models. In addition, we used the types of models most common in functional connectivity brain-phenotype studies[51]. Yet, complex models like neural networks are likely more susceptible to leakage due to their ability to memorize data[52,53]. Relatedly, many other evaluation metrics exist, such as mean squared error and mean absolute error; we focused primarily on $r$ and secondarily on $q^2$ because $r$ is the most common performance metric in neuroimaging trait prediction studies[51].

Another limitation is that leakage is not always as well-defined as in this paper. Some examples are universally leakage, such as ignoring family structure, accidentally duplicating data, and selecting features in the combined training and test data. In other cases, whether training and test data are independent may depend on the goal. For example, one may wish to develop a model that will be applied to data from a new site, in which case leave-one-site-out prediction would be necessary. Here, leakage would be present if data from the test site were included when training the model. However, other applications, such as those presented in this paper, may not require data to be separated by site and can instead apply site correction methods. Similarly, if one wishes to demonstrate that a model generalizes across diagnostic groups, the model should be built on one group and tested on another. The application-dependent nature of leakage highlights the importance of attention to detail and thoughtful experimentation in avoiding leakage.

Concerns about reproducibility in machine learning[17] can be partially attributed to leakage. As expected, feature and subject leakage inflated prediction performance. Positively, many forms of leakage did not exhibit inflated results. Additionally, larger samples and running multiple train and test splits mitigated inflated results. Since the effects of leakage are wide-varying and not known beforehand, the best practice remains to be vigilant and avoid data leakage altogether.

## Methods
### Preprocessing
In all datasets, data were motion corrected. Additional preprocessing steps were performed using BioImage Suite[54]. This included regression of covariates of no interest from the functional data, including linear and quadratic drifts, mean cerebrospinal fluid signal, mean white matter signal, and mean global signal. Additional motion control was applied by regressing a 24-parameter motion model, which included six rigid body motion parameters, six temporal derivatives, and the square of these terms, from the data. Subsequently, we applied temporal smoothing with a Gaussian filter (approximate cutoff frequency=0.12 Hz) and gray matter masking, as defined in common space. Then, the Shen 268-node atlas[55] was applied to parcellate the denoised data into 268 nodes. Finally, we generated functional connectivity matrices by correlating each pair of node time series data and applying the Fisher transform.

Data were excluded for poor data quality, missing nodes due to lack of full brain coverage, high motion (>0.2 mm mean frame-wise motion), or missing behavioral/phenotypic data, which is detailed further for each specific dataset below.

### Adolescent Brain Cognitive Development Data
In this work, we used the first and second releases of the first year of the ABCD dataset. This consists of 9-10-year-olds imaged across 21 sites in the United States. 7970 participants with resting-state connectomes (up to 10 minutes of resting-state data) remained in this dataset after excluding scans for poor quality or high motion (>0.2 mm mean frame-wise displacement [FD]). Family information was not

available for one participant, leaving 7969 participants, with 6903 unique families. Among these participants, the mean age was 9.94 (s.d. 0.62) years, with a range of 9-10.92 years, and 49.71% self-reported their sex as female.

For the attention problems measure, we used the Child Behavior Checklist (CBCL)[56] Attention Problems Raw Score. One participant was missing the attention problems score in ABCD. Of the participants with an attention problems score, the mean was 2.80 (s.d. 3.40), with a range of 0-20.

For the matrix reasoning measure, we used the Wechsler Intelligence Scale for Children (WISC-V)[57] Matrix Reasoning Total Raw Score. WISC-V measurements were missing from 147 participants ($N = 7822$). The mean matrix reasoning score in ABCD was 18.25 (s.d. 3.69), with a range of 0-30.

### Healthy Brain Network Data
The HBN dataset consists of participants from approximately 5-22 years old. The data were collected from four sites near the New York greater metropolitan area. 1201 participants with resting-state connectomes (10 minute scans) remained after applying the exclusion criteria. 39.80% were female, and the average age was 11.65 (s.d. 3.42) years, with a range of 5.58–21.90. Family information is not available in this dataset.

For the attention problems measure, we used the CBCL Attention Problems Raw Score[56]. 51 participants were missing the attention problems score, but the mean score of the remaining participants was 7.41 (s.d. 4.54), with a range of 0–19.

For the matrix reasoning measure, we also used the WISC-V[57] Matrix Reasoning Total Raw Score. 177 participants were excluded for missing this measure. The mean score was 18.36 (s.d. 4.46), with a range of 2–31.

### Human Connectome Project Development Data
The HCPD dataset includes healthy participants ages 8–22, with imaging data acquired at four sites across the United States (Harvard, UCLA, University of Minnesota, Washington University in St. Louis). 605 participants with resting-state connectomes (up to 26 min of resting-state data) remained after excluding low-quality or high-motion data. Among these 605 participants, the average age was 14.61 (s.d. 3.90) years, ranging from 8.08 to 21.92 years. 53.72% of participants self-reported their sex as female, and there were 536 unique families.

For the attention problems measure, we used the Child Behavior Checklist (CBCL)[56] Attention Problems Raw Score. 462 participants had this measure available, and the mean was 2.03 (s.d. 2.56), with a range of 0–18.

For the matrix reasoning measure, we used the WISC-V[57] Matrix Reasoning Total Raw Score. 424 participants remained in this analysis, with a mean of 21.08 (s.d. 3.96) and a range of 11–31.

### Philadelphia Neurodevelopmental Cohort Data
The PNC dataset consists of 8-21 year-olds in the Philadelphia area who received care at the Children's Hospital of Philadelphia. 1126 participants with resting-state scans (six minute scans) passed our exclusion criteria. The average age was 14.80 (s.d. 3.29) years, with a range of 8–21. The percentage of self-reported female participants was 54.62%.

For the attention problems measure, we used the Structured Interview for Prodromal Symptoms[58]: Trouble with Focus and Attention Severity Scale (SIP001, accession code: phv00194672.v2.p2). 1104 participants had this measure available, and the mean was 1.03 (s.d. 1.19), with a range of 0–6.

For the matrix reasoning measure, we used the Penn Matrix Reasoning[59,60] Total Raw Score (PMAT_CR, accession code: phv00194834.v2.p2). 1119 participants remained in this analysis, with a mean of 11.99 (s.d. 4.09) and a range of 0–24.

## Baseline models

For the primary analyses, we trained a ridge regression model using 5-fold cross-validation[27]. For HBN, HCPD, and PNC, five nested folds were used for hyperparameter selection, while only two nested folds were used in ABCD to reduce computation time. Within the folds, the top 5% of features most significantly correlated with the phenotypic variable were selected. Furthermore, we performed a grid search over the L2 regularization parameter $\alpha$ ($\alpha = 10^{\{-3,-2,-1,0,1,2,3\}}$), with the chosen model being the one with the highest Pearson's correlation value $r$ in the nested folds.

For our baseline gold standard model (Figs. 2–5: labeled as gold standard), the data were split accounting for family structure, where applicable (ABCD and HCPD only), such that all members of a single family were included in the same test split. In addition, we performed cross-validated covariate regression, where several covariates were regressed from the functional connectivity data within the cross-validation scheme[38,40]. Covariates were first regressed from the training data, and then those parameters were applied to regress covariates from the test data. The covariates included mean head motion (FD), sex, and age, though age was not regressed from the data for models predicting age. Furthermore, where applicable (ABCD, HBN, and HCPD), site differences were corrected within the cross-validation scheme using ComBat[29–31]. ComBat was performed separately from covariate regression because ComBat is designed for batch effects, not continuous variables[61]. In addition to the baseline gold standard model, we evaluated numerous forms of leakage, as described in the following sections (see also Fig. 1).

## Selection of forms of leakage

Due to the myriad forms of leakage, investigating every type of leakage is not feasible. In this work, we focused on three broad categories of leakage, including feature selection leakage, covariate-related leakage (leakage via site correction or covariate regression), and subject-level leakage (leakage via family structure or repeated subjects). We chose these particular forms of leakage because we expect that they are the most common and/or impactful errors in neuroimaging prediction studies. In our experience, feature selection leakage is an important consideration because it may manifest in subtle ways. For example, one may perform an explanatory analysis, such as determining the most significantly different brain networks between two groups, and then use these predetermined networks as predictive features, which constitutes leakage. As for covariate-related leakage, we have noticed that site correction and covariate regression are often performed on the combined training and test data in neuroimaging studies. Finally, for subject-level leakage, family structure is often ignored in neuroimaging datasets, unless it is being explicitly studied. Thus, understanding how prediction performance may be altered by these forms of leakage remains an important question.

Some forms of leakage were not considered in this work, including temporal leakage, the selection of model hyperparameters in the combined training/test data, unsupervised dimensionality reduction in the combined training/test data, standardization of the phenotype, and illegitimate features. Temporal leakage, where a model makes a prediction about the future but uses test data from a time point prior to the training data[17], is not relevant for cross-sectional studies using static functional and structural connectivity, but it could be relevant for prediction studies using longitudinal data or brain dynamics. While evaluating models in the test dataset to select the best model hyperparameters is a form of leakage, it has been previously studied in neuroimaging[49] and therefore was not included in this study. Various forms of unsupervised dimensionality reduction, such as independent component analysis[62,63], are popular in neuroimaging and constitute leakage if performed in the combined training and test datasets prior to performing prediction. Leakage via unsupervised dimensionality reduction should be avoided, but we felt that leakage via feature selection is a

dimensionality reduction technique that is both more common and more impactful, due to its involvement of the target variable. Moreover, phenotypes are sometimes standardized (i.e., z-scored) outside of the cross-validation folds, but this form of leakage was not investigated in this work because it is insensitive to the most common evaluation metrics in prediction using neuroimaging connectivity (Pearson's $r$, $q^2$). Finally, leakage via illegitimate features entails the model having access to features which it should not, such as a predictor that is a proxy for the outcome variable[17]. Studies using both imaging and clinical or phenotypic measures to predict other outcomes should be cautious of leakage via illegitimate features, but it is less relevant for studies predicting strictly from structural or functional connectivity data.

## Feature leakage

Feature selection is a common step in many connectome-based machine learning pipelines. Often, it consists of determining which features are the most relevant to the phenotype of interest, such as by selecting the edge features most correlated with this phenotype. A possible mistake is selecting features in the entire dataset, then training models and evaluating performance with k-fold cross-validation or a train/test split. Whelan and Garavan[64] previously demonstrated inflated performance via non-nested feature selection using random data, though we think it is also useful to demonstrate this with neuroimaging data. To do so, we selected the top 5% of features in the entire dataset, and then evaluated the performance using 5-fold cross-validation.

## Covariate-related leakage

A common example of leakage is correcting for site effects, such as with ComBat[29–31], in the whole dataset prior to splitting data into training and test sets. To avoid leakage, one should apply ComBat to the training data within each fold of cross-validation, and then use those ComBat parameters to correct for the site of the test data, as described in a recent work by Spisak[61]. Here, we evaluated the effect of leakage via performing ComBat on the entire dataset and compared this to the gold standard of applying ComBat within cross-validation folds. As described in several previous works[29–31], ComBat performs the correction based on the following equation for feature $v$ of the data $X$, with site $i$ and scan $j$[29–31]:

$$x_{i,j,v}^{adjusted} = \frac{x_{i,j,v} - \hat{\alpha}_v - \hat{\gamma}_{i,v}^*}{\delta_{i,v}^*} + \hat{\alpha}_v \tag{1}$$

where $\alpha_v$ is the overall measure for that edge, $\gamma_{i,v}$ is the additive site effects for site $i$ and feature $v$, and $\delta_{i,v}$ is the multiplicative site effects for site $i$ and feature $v$[30]. However, unless performed in a cross-validated manner, leakage occurs when estimating $\alpha_v$, $\gamma_{i,v}$, and $\delta_{i,v}$ on the combined training and test data. In practice, these parameters should be estimated only in the training data, and then applied to correct for site in both the training and test data. Crucially, we are performing site correction within individual datasets, where the imaging parameters are generally harmonized across sites. As such, we may underestimate the effects of site in other scenarios, such as when combining small datasets into one large study.

Another common form of leakage is covariate regression. For example, regressing the covariates/confounds out of the whole dataset in a non-cross-validated manner has been shown to negatively affect prediction performance[40]. The following equations describe regressing covariates out of the whole dataset for feature $X$ of size [$N$ x $p$], covariates $C$ of size [N x (k + 1)], which includes an intercept term, and the OLS solution $\hat{\beta}$[38]:

$$\hat{\beta} = \left(C^T C\right)^{-1} C^T X \tag{2}$$

$$X_{adjusted} = X - C\hat{\beta} \tag{3}$$

Leakage occurs above by finding the OLS solution $\hat{\beta}$ using both training and test data. Performing covariate regression within the cross-validation folds is the recommended solution to avoid leakage[38,40]:

$$\hat{\beta}_{train} = (C_{train}^T C_{train})^{-1} C_{train}^T X_{train} \qquad (4)$$

$$X_{train,adjusted} = X_{train} - C_{train}\hat{\beta}_{train} \qquad (5)$$

$$X_{test,adjusted} = X_{test} - C_{test}\hat{\beta}_{train} \qquad (6)$$

The above equations avoid leakage by finding the OLS solution $\hat{\beta}$ only in the training data, and subsequently applying it to the test data. Cross-validated and non-cross-validated covariate regression have been compared in several previous studies[38,40], and we build upon those previous works here by evaluating them in three phenotypes and four datasets through the view of leakage.

### Subject-level leakage
Neuroimaging datasets, such as ABCD and HCPD, are often oversampled for twins or siblings. Given the heritability of brain structure and function[32–34], family structure should be specifically accounted for in analysis pipelines, such as in permutation tests[65]. In the context of this work, having one family member in the training set and another in the test split is a form of leakage. For instance, in a hypothetical case of predicting a highly heritable phenotype from a highly heritable brain network, the model could memorize the data from one family member in the training set and strongly predict the phenotype of another family member in the test set. However, splitting the data into cross-validation folds by family instead of on the individual subject level prevents leakage. If a dataset contains 1000 participants from 500 unique families, 5-fold cross-validation should include training data belonging to 400 randomly selected families, and test data belonging to the other 100 randomly selected families.

Another possible case of leakage is where various scans from the same participant are treated as separate samples, which we call subject leakage. This could include, for instance, treating longitudinal data as cross-sectional or including various runs (or tasks) of fMRI acquisition for the same participant as separate data points. To evaluate an extreme version of subject leakage, we considered the case when a certain percentage of the subjects in the dataset were repeated. For example, in a hypothetical dataset with 1000 participants, 20% subject leakage would include an additional 200 repeats of random participants in the dataset, for a total dataset size of 1200. Then, in this larger sample, we repeated nested cross-validation and compared the prediction performance results to the original sample. In this form, subject leakage is not directly comparable to longitudinal or repeated measurements studies, since we are instead duplicating the exact same scan. Yet, we can use repeated subjects to demonstrate the concept of leakage. Notably, we did not account for family structure in subject leakage, otherwise the leaked subjects would always be in the same training or test split.

### Evaluation metrics
Our main evaluation metric is Pearson's correlation $r$ between the true and predicted phenotype. This evaluation metric does not necessarily reflect predictive power but is commonly used to establish brain-behavior relationships[28]. In addition, we reported cross-validation $R^2$, also called $q^2$, as defined below[28]:

$$q^2 = 1 - \frac{\Sigma(y - y_{pred})^2}{\Sigma(y - \bar{y})^2} \qquad (7)$$

where $y$ and $y_{pred}$ are the observed and predicted behavior, respectively, and $\bar{y}$ is the mean of the observed behavior. Performance metrics were calculated by concatenating predictions across folds, and then these metrics were calculated for 100 random seeds of cross-validation. Importantly, $q^2$ is sometimes substantially negative when the model prediction gives a higher mean-squared error than predicting the mean.

### Sample size
To evaluate the interaction between sample size and leakage, we randomly sampled without replacement $N = 100, 200, 300$, and 400 participants from ABCD, HBN, HCPD, and HBN. The ABCD data was only resampled from the four largest sites (total $N = 2436$) to avoid issues with ComBat where only one data point was present from a particular site after resampling. In addition, for the two datasets with family structure (ABCD and HCPD), data were resampled by family, not by individual, to keep approximately the same proportion of related participants in our subsample. We repeated the resampling procedure 10 times for each dataset and sample size, and for each resampled dataset we evaluated the gold standard and leaky prediction performance across 10 repeats of 5-fold nested cross-validation.

### Structural connectome analysis
Our structural connectome analyses included 635 participants from the HCPD dataset. We started with the diffusion tensor data of the 635 participants. Then, we corrected for susceptibility artifacts and applied DSI-studio to reconstruct the diffusion data using generalized q-sampling imaging. Finally, we created structural connectomes using automatic fiber tracking for the Shen 268-node atlas[55].

### Additional models
We evaluated the effects of leakage in two additional models: support vector regression (SVR)[27] and connectome-based predictive modeling (CPM)[2]. In both cases, we performed 5% feature selection, as described above. For SVR, the radial basis function was used, and we performed a grid search for the L2 regularization parameter ($C = 10^{[-3,-2,-1,0,1,2,3]}$), where $C$ is inversely proportional to regularization strength. For CPM, the positive and negative features were combined into a single number, and a univariate linear regression model was then fitted.

### Reporting summary
Further information on research design is available in the Nature Portfolio Reporting Summary linked to this article.

## Data availability
Data are available through the Adolescent Brain Cognitive Development Study[22] (https://abcdstudy.org/), the Healthy Brain Network Dataset[23] (https://data.healthybrainnetwork.org/main.php), the Human Connectome Project Development Dataset[24] (https://www.humanconnectome.org/study/hcp-lifespan-development/overview), and the Philadelphia Neurodevelopmental Cohort Dataset[25,26] (https://www.med.upenn.edu/bbl/philadelphianeurodevelopmentalcohort.html, accession code: phs000607.v3.p2). The ABCD dataset was downloaded from the NIMH Data Archive (NDA). The HBN dataset was downloaded via the HBN portal on through the Longitudinal Online Research and Imaging System (LORIS). The HCP-Development 2.0 Release dataset was downloaded from the NDA and came from https://doi.org/10.15154/1520708. The PNC dataset was downloaded via dbGaP (accession code: phs000607.v3.p2). Funding details of these datasets are included in the Acknowledgments section of the manuscript. Source data (i.e., prediction performance values) that can be used for creating the plots are provided with this work and also available at https://github.com/mattrosenblatt7/leakage_neuroimaging. Source data are provided with this paper.

## Code availability

Preprocessing was carried out using BioImage Suite, which is freely available here: (https://medicine.yale.edu/bioimaging/suite/). Code for the analyses and plots is available at: https://github.com/mattrosenblatt7/leakage_neuroimaging[66]. This includes a link to a Google Colaboratory session, where an environment is set up to re-create all plots (based on a processed.csv file of the results from our scripts). Please note that the connectome files cannot be shared due to data sharing restrictions, and thus we do not provide data for running the main analysis code. Python 3.7 was used to analyze the data, and additional packages included numpy 1.24.3[67], pandas 2.0.3[68], scikit-learn 1.2.2[27], and scipy 1.10.1[69].

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

## Acknowledgements

This study was supported by the National Institute of Mental Health grant R01MH121095 (obtained by D.S.). M.R. was supported by the National Science Foundation Graduate Research Fellowship under grant DGE2139841. L.T. was supported by the Gruber Science Fellowship. S.N. was supported by the National Institute of Mental Health under grant R00MH130894. Any opinions, findings, and conclusions or recommendations expressed in this material are those of the authors and do not necessarily reflect those of the funding agencies. We have a Yale IRB exception (HIC: 2000023326) to use open-source neuroimaging data. The Human Connectome Project Development data was supported by the National Institute Of Mental Health of the National Institutes of Health under Award Number U01MH109589 and by funds provided by the McDonnell Center for Systems Neuroscience at Washington University in St. Louis. The HCP-Development 2.0 Release data used in this report came from https://doi.org/10.15154/1520708. Additional data used in the preparation of this article were obtained from the Adolescent Brain Cognitive DevelopmentSM (ABCD) Study (https://abcdstudy.org), held in the NIMH Data Archive (NDA). This is a multisite, longitudinal study designed to recruit more than 10,000 children age 9–10 and follow them over 10 years into early adulthood. The ABCD Study® is supported by the National Institutes of Health and additional federal partners under award numbers U01DA041048, U01DA050989, U01DA051016, U01DA041022, U01DA051018, U01DA051037, U01DA050987, U01DA041174, U01DA041106, U01DA041117, U01DA041028, U01DA041134, U01DA050988, U01DA051039, U01DA041156, U01DA041025, U01DA041120, U01DA051038, U01DA041148, U01DA041093, U01DA041089, U24DA041123, and U24DA041147. A full list of supporters is available at https://abcdstudy.org/federal-partners.html. A listing of participating sites and a complete listing of the study investigators can be found at https://abcdstudy.org/consortium_members/. ABCD consortium investigators designed and implemented the study and/or provided data but did not necessarily participate in the analysis or writing of this report. This manuscript reflects the views of the authors and may not reflect the opinions or views of the NIH or ABCD consortium investigators. The Healthy Brain Network (http://www.healthybrainnetwork.org) and its initiatives are supported by philanthropic contributions from the following individuals, foundations and organizations: Margaret Bilotti; Brooklyn Nets; Agapi and Bruce Burkard; James Chang; Phyllis Green and Randolph Cōwen; Grieve Family Fund; Susan Miller and Byron Grote; Sarah and Geoff Gund; George Hall; Jonathan M. Harris Family Foundation; Joseph P. Healey; The Hearst Foundations; Eve and Ross Jaffe; Howard & Irene Levine Family Foundation; Rachael and Marshall Levine; George and Nitzia Logothetis; Christine and Richard Mack; Julie Minskoff; Valerie Mnuchin; Morgan Stanley Foundation; Amy and John Phelan; Roberts Family Foundation; Jim and Linda Robinson Foundation, Inc.; The Schaps Family; Zibby Schwarzman; Abigail Pogrebin and David Shapiro; Stavros Niarchos Foundation; Preethi Krishna and Ram Sundaram; Amy and John Weinberg; Donors to the 2013 Child Advocacy Award Dinner Auction; Donors to the 2012 Brant Art Auction. Additional data were provided by the PNC (principal investigators Hakon Hakonarson and Raquel Gur; phs000607.v1.p1). Support for the collection of these datasets was provided by grant RC2MH089983 awarded to Raquel Gur and RC2MH089924 awarded to Hakon Hakonarson.

## Author contributions

M.R. and D.S. conceptualized the study. L.T. and D.S. curated the data. M.R. performed the formal analysis. M.R. and D.S. drafted the manuscript. L.T., R.J., S.N., and D.S. reviewed and edited the manuscript. M.R., R.J., S.N., and D.S. contributed to the visualizations. D.S. supervised the project.

## Competing interests

The authors declare no competing interests.

## Ethical approval

The four datasets used in this study were each supervised by their relevant ethical review boards.

## Informed consent

Informed consent was obtained by the relevant data collection teams in the four public datasets.
