## [Peer Review File · Nature Communications]

Data leakage inflates prediction performance in connectome-based machine learning modelsReviewer #1 (Remarks to the Author):

The paper describes the effects of data leakage in predictive models in the field of neuroimaging. The authors consider multiple datasets, multiple forms of leakage, and multiple model types. In general, they find that some forms of leakage inflate predictive performance, sometimes drastically.

The findings are a valuable reminder of the importance of avoiding leakage. They also help identify types of leakage that are particularly harmful. Leakage is never a correct practice, as the authors emphasize. Still, given that leakage in published work is already widespread, it will be important to know which previously reported results are likely to be substantially impacted, and these findings help toward that goal.

I have a few suggestions for improvement.

* Given the wide applicability of predictive modeling techniques, what (if anything) can readers outside neuroimaging take away from this paper? The answers will necessarily be somewhat speculative, but I believe it is appropriate to include a discussion, especially given the multidisciplinary nature of the journal.

* The only metrics used are r and R^2/q^2 . This is a slight limitation which should be acknowledged. Notably, these metrics are unable to provide evidence of leakage arising from leaky z-scoring. (The authors note this about r but I think it's true of R^2 as well.)

* For this reason, I strongly suggest removing z-scoring from the paper. Since you have used metrics that are mathematically guaranteed to be insensitive to it, it is equivalent to not having performed a z-scoring analysis. As it is, the figures in the paper give a misleading impression. Alternatively, measure the effect of z-scoring using a metric such as mean squared error.

* I think "1 -" is missing in the definition of q^2 .

* One of the q^2 values is -0.13, which I think merits an explanation. While I understand that the value can theoretically be negative, I think it's unlikely to happen in practice. Also, if it's -0.01 it won't raise any eyebrows, but -0.13 is quite substantial.

* The findings on family leakage are unconvincing. In both datasets with family structure, only a small % of participants belong to families with more than one individual. So it is entirely possible that family leakage has an effect that you weren't able to detect. I would strongly recommend performing an additional analysis with a sample that's limited to families with more than one individual.

* Some additional information would be helpful to contextualize this finding. (This goes back to my earlier comment about helping the reader understand the extent to which the findings are specific to the datasets / target variables studied in the paper.) What is the heritability of the phenotypes in question? If those are low, perhaps that can account for the insignificance of family leakage (assuming it is still insignificant on a sample limited to families). Alternatively, perhaps the model doesn't have the capacity to memorize the data. Kapoor and Narayanan (17) point out that in a civil war prediction modeling task, a random forests model was much better at learning spurious patterns, and hence much more susceptible to leakage, compared to logistic regression. The use of SVR / RBF (which I understand to be nonlinear) in this paper should make this possibility less likely, but then again, maybe 400 samples is not sufficient to learn the patterns that would allow the model to exploit family structure.

* Considering that the paper is about reproducibility, the authors should make their analysis code available.

This review was authored by Arvind Narayanan.

Reviewer #2 (Remarks to the Author):

In this paper, Rosenblatt et al., focus on the implications of different types of the common problem of data leakage. They systematically test three broad types of data leakage with connectivity-based predictions, including feature selection, covariate correction, and dependency between participants, with four different open datasets and three different predicted measures. This is an important evaluation of the impact of common errors in neuroimaging predictive modeling and will inform the field in minimizing the risks of data leakage affecting predictive performance and reproducibility. The paper is clear and well written, and I think it should be accepted for publication following a relatively minor revision. My main comments are:

1. The paper focuses on connectivity-based predictions (mostly functional, and one model with structural connectivity). This should be made explicit in the abstract, and potentially (but not necessarily) also in the title, since different types of predictive modeling (e.g., based on task fMRI) may show different patterns.
2. The paper nicely illustrates the problem of data leakage and when its effect is larger. However, I think the discussion of potential solutions is somewhat lacking. The authors focus on suggesting sharing the analysis code so that other researchers (or reviewers) could reproduce the result and test for potential leakage. While I completely agree that this is a very good practice, I doubt it will solve the issue, since even publicly shared analysis codes are rarely inspected by others. The authors do briefly mention the model checklists suggested by a previous paper (Kapoor and Narayanan, 2022), which might help authors, reviewers and editors identify leakage issues. I recommend extending the discussion of solutions a bit, by describing the suggested solutions more carefully and, if possible, suggesting additional ones (e.g., clear guidelines on how to avoid data leakage).

Minor:

1. The length of the resting-state data was different across datasets (26 minutes for HCPD, 10 minutes for ABCD and HBN, and 6 minutes for PNC). Could this difference affect the results across the four datasets?
2. Line 14 page 13: "Excluding covariate regression (median rcoef=0.31-0.86), site correction (median rcoef=0.86-0.99), or both (median rcoef=0.32-0.82) resulted in moderate changes in the coefficients." This sentence sounds like either excluding covariate regression or excluding site correction had similar effects on the model coefficients, while in fact the effect of excluding covariate regression was much stronger (the ranges of medians do not even overlap). This should be more clearly stated.
3. Page 3 line 37: I believe there's a typo, and $N = 4059$ should be $N = 4079$.
4. Figure 2 caption: "Rows represent different leakage types" this is an error since this is a figure of non-leaky analysis choices, and rows here figure represent different analysis choices.

Reviewer #3 (Remarks to the Author):

Many publications that use machine learning report an inflated prediction performance. One of the causes is an incorrect validation scheme, in which the training and testing data are not independent and thus the test data is not really unseen -- information "leaks" into the model training.

In this study, the authors conduct extensive experiments to measure the impact of such incorrect validation schemes on datasets and prediction tasks that have been widely used in the neuroimaging literature. Several typical mistakes in the validation strategy are applied on combinations of 3 prediction targets and 4 datasets subsampled at 4 different sample sizes.

The authors explore 3 classic families of mistakes: fitting part of the pipeline on the full dataset (2.8), errors related to covariate regression (2.9), failing to account for dependencies between data points (2.10).

Some other common errors such as dimensionality reduction on the whole dataset or hyperparameter overfitting are not considered.

The worse cases of prediction performance over-estimation occur for fitting feature selection on the whole data ("feature leakage") and overlapping train and test sets ("subject leakage"). Those results are expected but it is useful to observe them on real neuroimaging data.

The paper is well written, the objectives and experiments are clear and convincing.

I would have liked to see more discussion on the following points:

The authors correctly write that including in the train set data points that are not independent from testing data points increases scores, and therefore we should be careful to define groups (for example families) that must not be spread between train and test. However the definition of such groups can depend on the particular task at hand. For example, if we want to evaluate a model that should generalize across sites, we should leave out whole sites in the cross validation. However, if the application we envision makes some data from the site where the model will be deployed available during training, not leaving out whole sites may be acceptable. Could the authors comment on what kind of dependencies can exist between data points (shared family, site, subject, ...) and on when we should take care to leave out whole groups?

Could the authors discuss a bit more the kinds of leakage they chose to leave out of this study, and how the considered mistakes were chosen: are they more common, more likely to cause inflated scores, etc. I do think the choice made in this study is appropriate because kinds of leakage that involve the target variable (as considered here), rather than only features (eg unsupervised dimensionality reduction on the whole dataset), are far worse.

Regarding site and covariate regression: I would argue that a better approach would be to not regress covariates out, but compare to baseline models that have only access to the covariates. I am not sure covariate regression is well-founded in a predictive setting or that we can talk about "leakage via illegitimate features" -- in all situations where a machine model would be used it would be possible to have subjects' sex and age. But if we want to check that brain imaging data brings additional information, we can compare to baselines that use only the covariates. For generalization across sites, rather than removing site effects we should evaluate performance on left-out sites. A more detailed discussion can be found in

Dockès, J., Varoquaux, G. and Poline, J.B., 2021. Preventing dataset shift from breaking machine-learning biomarkers. *GigaScience*, 10(9), p.giab055.

Regarding the recommendations to avoid leakage in future studies: studies that are prone to leakage are likely the same studies that are prone to having code that is complex, poorly written and difficult to understand, and unlikely to be reused. Therefore, I wonder how likely cross-validation errors in this code would be discovered by reviewers or others, even if the code was made public. Could the authors provide a few more suggestions, perhaps some sanity checks with synthetic data, the use of well-maintained packages for

implementing the cross-validation, etc.?

REVIEWER COMMENTS

We thank all the reviewers for their thoughtful comments. Below, we have included a point-by-point response. Updated text is shown in blue. References referred to in this response can be found at the end of the document.

We also want to bring attention to three general changes in this work.

First, we fixed some errors in our processed ABCD dataset that alter some of the prediction performance results, but do not change any conclusions. We noticed an indexing error, where we loaded some of the incorrect connectivity matrices in ABCD. We have fixed this error, as well as included an additional 3820 participants from the “Wave 2” release of ABCD, which we recently finished processing. We found that prediction performance increased in ABCD, likely due to both the fixed connectomes and the additional sample size, but none of the conclusions regarding leakage were changed. Figures 7, 8, 9, S5-S13 have been updated with the new results.

Second, based on a recommendation from a colleague, we changed the phenotype “attention” throughout this work to “attention problems.”

Third, based on the editorial formatting guide, we moved the methods section to after the discussion section.

Reviewer #1 (Remarks to the Author):

The paper describes the effects of data leakage in predictive models in the field of neuroimaging. The authors consider multiple datasets, multiple forms of leakage, and multiple model types. In general, they find that some forms of leakage inflate predictive performance, sometimes drastically.

The findings are a valuable reminder of the importance of avoiding leakage. They also help identify types of leakage that are particularly harmful. Leakage is never a correct practice, as the authors emphasize. Still, given that leakage in published work is already widespread, it will be important to know which previously reported results are likely to be substantially impacted, and these findings help toward that goal.

I have a few suggestions for improvement.

Thank you for your time reviewing this paper and thoughtful suggestions. We have incorporated several improvements, which are detailed below.

1. Given the wide applicability of predictive modeling techniques, what (if anything) can readers outside neuroimaging take away from this paper? The answers will necessarily be somewhat speculative, but I believe it is appropriate to include a discussion, especially given the multidisciplinary nature of the journal.

We agree that extending the discussion to include general takeaways will be valuable to the manuscript. We first describe the benefits of quantifying the effects of leakage in a specific field and then call on researchers in other fields to do so. Then, we detail several strategies to prevent leakage. Please see the following addition to the discussion section (page 15, line 43):

“The results presented in this work focus on neuroimaging, specifically functional and structural connectivity prediction studies. However, the lessons from this work may be valuable to any field using scientific machine learning. Since we expect that leakage will be prevalent across many fields for the foreseeable future, quantifying the effects of leakage may provide valuable field-specific insight into the validity of published results. Thus, we encourage researchers in other fields to use their domain-specific knowledge to identify possible forms of leakage and subsequently evaluate their effects. Although quantifying the effects is essential due to the pervasiveness of leakage, researchers should pay careful attention to avoiding leakage.

Numerous strategies can help prevent leakage in neuroimaging and other machine-learning applications. These strategies include carefully developing and sharing code, alternative validation strategies, model information sheets (Kapoor and Narayanan, 2023), skepticism about one’s own results, and cross-disciplinary collaborations. Writing and maintaining code should incorporate several facets to reduce the likelihood of leakage, including establishing an analysis plan prior to writing code, using well-maintained packages, and sharing code. One’s analysis plan should be set ahead of time, either informally or, if appropriate, formally through pre-registration. As one tries more pipelines, especially if searching for a significant result (i.e., “p-hacking”), leakage is more likely to occur. A predefined plan could minimize the likelihood of leakage by detailing how features will be selected, which models will be trained, and how possible covariates and nested structures will be handled. Another suggestion for reducing the likelihood of leakage is using well-maintained packages. For example, Scikit-learn has a k-fold cross-validation package (Pedregosa et al., 2011) that has been thoroughly tested, whereas developing k-fold cross-validation code from scratch may lead to accidental leakage. Among

many other benefits, sharing code, particularly well-documented code, could decrease the effects of leakage by allowing external reviewers to investigate published pipelines for leakage. Relatedly, although not always possible, distributing preprocessed data can make the reproduction of results much easier and less time-consuming for reviewers or those who want to verify the validity of a predictive model.

Moreover, most neuroimaging papers are evaluated with train/test splits or k-fold cross-validation. However, alternative validation strategies, such as a lockbox (Hosseini et al., 2020) or external validation, may reduce the likelihood of leakage. Both these strategies help to maintain a clearer separation between training and test data, where a lock box entails leaving out a subset of the data until a final evaluation (Hosseini et al., 2020) and external validation consists of applying a model to a different dataset. Another strategy to decrease the prevalence of leakage is using model information sheets, such as the one proposed by Kapoor and Narayanan (Kapoor and Narayanan, 2023). Model information sheets allow for the authors, reviewers, and public to reflect upon the work and identify possible leakage. However, it may be difficult to verify the accuracy of model information sheets when data cannot be shared (Kapoor and Narayanan, 2023). This limitation is especially true for neuroimaging datasets, which often require applications to access the data. As a result, we also recommend healthy skepticism of one's results. For instance, if a machine learning pipeline leads to a surprising result, the code should be scrutinized by asking a collaborator to view one's code or repeat the analyses on synthetic data. Finally, collaborations across disciplines to incorporate domain and machine learning experts will help prevent leakage (Kapoor and Narayanan, 2023). Domain experts can bring knowledge of the nuances of datasets (e.g., the prevalence of family structures in neuroimaging datasets). In contrast, machine learning experts can help domain experts train models to avoid leakage."

2. The only metrics used are r and R^2/q^2 . This is a slight limitation which should be acknowledged. Notably, these metrics are unable to provide evidence of leakage arising from leaky z-scoring. (The authors note this about r but I think it's true of R^2 as well.)

We have removed leaky z-scoring from the paper (see next point) and have also added the following to the limitations section (page 17, line 15):

"Relatedly, many other evaluation metrics exist, such as mean squared error and mean absolute error; we focused primarily on \$r\$ and secondarily on \$q^2\$ because \$r\$ is the most common performance metric in neuroimaging trait prediction studies (Yeung et al., 2022)."

** For this reason, I strongly suggest removing z-scoring from the paper. Since you have used metrics that are mathematically guaranteed to be insensitive to it, it is equivalent to not having performed a z-scoring analysis. As it is, the figures in the paper give a misleading impression. Alternatively, measure the effect of z-scoring using a metric such as mean squared error.*

We have followed your suggestion of removing leaky z-scoring from the paper, and have added a brief statement about this in a new paragraph about our selection of leakage types (page 21, line 5).

"Moreover, phenotypes are sometimes standardized (i.e., z-scored) outside of the cross-validation folds, but this form of leakage was not investigated in this work because it is insensitive to the most common evaluation metrics in prediction using neuroimaging connectivity (Pearson's \$r\$, \$q^2\$ )."

3. I think "1 -" is missing in the definition of q^2 .

Thank you for catching this mistake. We have updated the text with the corrected definition:

$$q^2 = 1 - \frac{\sum (y - y_{pred})^2}{\sum (y - \bar{y})^2}$$

4. One of the q^2 values is -0.13, which I think merits an explanation. While I understand that the value can theoretically be negative, I think it's unlikely to happen in practice. Also, if it's -0.01 it won't raise any eyebrows, but -0.13 is quite substantial.

Thank you for raising this point. In our experience, we have seen q^2 values that are quite negative for brain-phenotype models, specifically when there is little to no signal. For example, Scheinost et al. showed in Figure 3 of their paper (Scheinost *et al.*, 2019) that low correlations tend to correspond to negative q^2 values. Some of the correlations we observed in this work are also lower than those presented in (Scheinost *et al.*, 2019), leading to the substantially negative q^2 values.

Please see an example scatter plot of the prediction of attention problems in the HCPD dataset below. We also made two changes to the text to highlight that q^2 may be negative.

First, in the results section, we added the following description (page 3, line 23):

“Notably, q^2 may be negative when the model prediction gives a higher mean-squared error than predicting the mean, as was the case for attention problems.”

Second, in the methods section (“Evaluation metrics”), we added the following sentence (page 23, line 18):

“Importantly, q^2 is sometimes substantially negative when the model prediction gives a higher mean-squared error than predicting the mean.”

5. The findings on family leakage are unconvincing. In both datasets with family structure, only a small % of participants belong to families with more than one individual. So it is entirely possible that family leakage has an effect that you weren't able to detect. I would strongly recommend performing an additional analysis with a sample that's limited to families with more than one individual.

* Some additional information would be helpful to contextualize this finding. (This goes back to my earlier comment about helping the reader understand the extent to which the findings are specific to the datasets / target variables studied in the paper.) What is the heritability of the phenotypes in question? If those are low, perhaps that can account for the insignificance of family leakage (assuming it is still insignificant on a sample limited to families). Alternatively, perhaps the model doesn't have the capacity to memorize the data. Kapoor and Narayanan (17) point out that in a civil war prediction modeling task, a random forests model was much better at learning spurious patterns, and hence much more susceptible to leakage, compared to logistic regression. The use of SVR / RBF (which I understand to be nonlinear) in this paper should make this possibility less likely, but then again, maybe 400 samples is not sufficient to learn the patterns that would allow the model to exploit family structure.

Thank you for the above suggestions. We agree that the family leakage results require more robust testing. We have addressed the above two points with two new analyses. First, we evaluated the effects of leakage in a twin subset of the ABCD dataset. This subset contains a total of $n=563$ pairs of twins ($n=1126$ participants total). We included four additional phenotypes as well as one additional model (random forest) to consider heritability and model memorization capacity. Second, we performed a simulation in ABCD where we varied the percentage of data coming from participants with another family member in the dataset. We found that your suspicion is correct: family leakage had a greater effect in the twin study, and increasing the percentage of participants belonging to a family with more than one member led to increases in the effects of leakage.

The following figure and description will be added to the main text (page 6, line 8):

2.5 Additional family leakage analyses in ABCD

Since the two datasets in this study with family information contained mostly participants without any other family members in the dataset (HCPD: 471/605, ABCD: 5868/7969 participants did not have family members), we performed several additional experiments to determine the effects of family leakage with a larger proportion of families. We used ABCD instead of HCPD for these experiments because ABCD has more families with multiple members in the dataset.

First, ABCD was restricted to only twins ($N=563$ pairs of twins, 1126 participants total), after which we performed 100 iterations of 5-fold cross-validation for all three phenotypes and model types. In one case, family structure was accounted for in the cross-validation splits. In another, the family structure was ignored, constituting leakage. Leakage in the twin dataset exhibited minor to moderate increases in prediction performance (Figure 6), unlike when using the entire dataset. The inflation was $\Delta r=0.04$ for age and $\Delta r=0.02$ matrix reasoning and attention problems.

We included several additional phenotypes and models to compare how leakage may affect twin studies (Figure S2), which showed similar results. The phenotype similarity between the twin pairs did not have a strong relationship with changes in performance due to leakage

(Figure S3). Furthermore, based on a simulation study, leakage effects increased with the percentage of participants belonging to a family with more than one individual (Figure S4).

Figure 6. Prediction performance in ABCD comparing the gold standard to twin/family leakage. The black bar represents the median performance of the “gold standard” models across random iterations, and the histograms show prediction performance across 100 iterations of 20-fold cross-validation. See also Figures S2-4.

The following additional analyses are included in a new supplementary section: S3. *Family leakage analysis*.

S3. *Family leakage analysis*

Beyond the three original phenotypes and models in this study, we considered several additional phenotypes from the Child Behavioral Checklist (CBCL) (Achenbach and Ruffle, 2000), as well as one additional model (Random Forest). The phenotypes were the Anxiety and Depression CBCL Syndrome Scale Raw Score (Anx/Dep), Aggressive CBCL Syndrome Scale Raw Score (Aggression), Internal CBCL Syndrome Scale Raw Score (Internal), and the External CBCL Syndrome Scale Raw Score (External). The Random Forest was considered in case the other models (ridge regression, SVR, CPM) had too low of a memorization capacity that would limit the effects of leakage. For the Random Forest (Pedregosa *et al.*, 2011), 10 estimators were used, and a grid search was performed varying the maximum depth (3, 5, 7, 9).

We found that, across all models and phenotypes, the median performance of twin leakage was greater than the median gold standard performance (Figure S2). The effects of leakage were greater for ridge regression and SVR compared to CPM and the Random Forest. Although not tested, family leakage may become increasingly impactful for complex models, such as neural networks, that are trained with larger datasets and more participants per family.

Figure S2. Comparison of prediction performance between the gold standard and twin leakage in the ABCD twin subset, related to Figure 6. Boxplot elements were defined as follows: the center line is the median across 100 random iterations; box limits are the upper and lower quartiles; whiskers are 1.5x the interquartile range; points are outliers. Four models (ridge regression, SVR, CPM, Random Forest) and seven phenotypes were included. In all cases, the median performance was higher for twin leakage compared to the gold standard. AP: Attention CBCL Syndrome Scale Raw Score; MR: WISC-V Matrix Reasoning Total Raw Score; Anx/Dep: Anxiety and Depression CBCL Syndrome Scale Raw Score; Aggression: Aggressive CBCL Syndrome Scale Raw Score; Internal: Internal CBCL Syndrome Scale Raw Score; External: External CBCL Syndrome Scale Raw Score.

We compared the similarity of each phenotype between twins and the corresponding increase in prediction performance in a leaky vs. non-leaky pipeline. As a metric of similarity, we took the ratio of the mean absolute error of the phenotype between each twin pair to the mean absolute error between the participant and all non-twin participants, and this ratio was averaged across all participants. The MAE ratio for participant p is defined as:

$$MAE\ Ratio = \frac{MAE(y_p, y_{p,twin})}{MAE(y_p, y_{p,non-twin})}$$

Thus, a value closer to 0 reflects greater similarity of that phenotype between twins. We chose to use phenotype similarity instead of literature estimates of heritability because a phenotype such as age is not necessarily “heritable,” but it is identical (or nearly identical due to different interview dates) for twins. Similarly, the CBCL measures were determined by a parent questionnaire and thus may reflect the tendencies of a parent in answering questions rather than explicit heritability of a trait.

The most similar phenotypes did not necessarily show greater leakage effects (Figure S3). This result could point toward the limited memorization capacity of the model, given the study design. For example, if there were many members per family, the model may more easily “memorize” the signature of family members, and the effects of leakage may be greater.

Figure S3. Comparison of prediction performance with twin leakage (y-axis) and the gold standard (x-axis) colored by phenotype similarity, related to Figure 6. An MAE Ratio closer to zero entails greater similarity between twins. The shape of each point indicates the phenotype.

Furthermore, we performed a simulation that altered the percentage of one-individual families in the dataset. To do this, we started with only families with multiple individuals, and then we added in random fractions of the participants without family members. In general, the effects of leakage increased as the fraction of participants coming from families with multiple members increased (Figure S4). However, the effects were still relatively small.

Figure S4. Simulation varying the percentage of participants with family members in the ABCD datasets, related to Figure 6. Ridge regression was performed for 100 random iterations with the datasets subsample consisting of 30, 40, 50, 60, 70, 80, 90, and 100% multi-participant families (100% is restricted to only participants with other family members). The error bars reflect the 2.5th and 97.5th percentiles of the 100 random iterations. AP: Attention Problems, MR: Matrix Reasoning.

Finally, we updated the discussion to reflect these results (page 14, line 43):

Before: “Notably, neither leaky site correction nor family leakage had effects on prediction performance.”

After: “Notably, leaky site correction **did not affect** prediction performance. **Family leakage had little to no effect when using the entire dataset due to the small percentage of participants belonging to families with more than one individual. Still, a twin subset and various simulations demonstrated that the effects of family leakage are more pronounced when datasets have a larger proportion of multi-member families.**”

6. Considering that the paper is about reproducibility, the authors should make their analysis code available.

We previously included a data and code availability statement after the references, but we now realize that this may make it difficult for readers to find the link to our code. To improve the visibility, we have moved it to before the references, as well as separated the statements into “Data availability” and “Code availability” (page 24, line 10).

Data availability

Data are available through the Adolescent Brain Cognitive Development Study (22), the Healthy Brain Network Dataset (23), the Human Connectome Project Development Dataset (24), and the Philadelphia Neurodevelopmental Cohort Dataset (25, 26). Source data (i.e., prediction performance values) that can be used for creating the plots are provided with this work and also available at https://github.com/mattrosenblatt7/leakage_neuroimaging.

Code availability

Code for the analyses and plots is available at:
https://github.com/mattrosenblatt7/leakage_neuroimaging.

This review was authored by Arvind Narayanan.

Reviewer #2 (Remarks to the Author):

In this paper, Rosenblatt et al., focus on the implications of different types of the common problem of data leakage. They systematically test three broad types of data leakage with connectivity-based predictions, including feature selection, covariate correction, and dependency between participants, with four different open datasets and three different predicted measures. This is an important evaluation of the impact of common errors in neuroimaging predictive modeling and will inform the field in minimizing the risks of data leakage affecting predictive performance and reproducibility. The paper is clear and well written, and I think it should be accepted for publication following a relatively minor revision. My main comments are:

Thank you for your thoughtful review and recommendations for improvement. We have addressed your concerns in a point-by-point response below.

1. The paper focuses on connectivity-based predictions (mostly functional, and one model with structural connectivity). This should be made explicit in the abstract, and potentially (but not necessarily) also in the title, since different types of predictive modeling (e.g., based on task fMRI) may show different patterns.

Thank you for raising this point. First, we updated the title as follows: “The effects of data leakage on connectome-based machine learning models”

In addition, although the beginning of our abstract starts with broad goals of neuroimaging, we updated a later portion of the abstract to explicitly state that we are using functional and structural connectivity.

Before: “Here, using over 500 different pipelines spanning four large neuroimaging datasets and three phenotypes, we evaluated six forms of leakage fitting into three broad categories: feature selection, covariate correction, and lack of independence between subjects.”

After (page 1, line 26): “Here, we investigated the effects of leakage on machine learning models in two common neuroimaging modalities, functional and structural connectomes. Using over 400 different pipelines spanning four large datasets and three phenotypes, we evaluated five forms of leakage fitting into three broad categories: feature selection, covariate correction, and lack of independence between subjects.”

2. The paper nicely illustrates the problem of data leakage and when its effect is larger. However, I think the discussion of potential solutions is somewhat lacking. The authors focus on suggesting sharing the analysis code so that other researchers (or reviewers) could reproduce the result and test for potential leakage. While I completely agree that this is a very good practice, I doubt it will solve the issue, since even publicly shared analysis codes are rarely inspected by others. The authors do briefly mention the model checklists suggested by a previous paper (Kapoor and Narayanan, 2022), which might help authors, reviewers and editors identify leakage issues. I recommend extending the discussion of solutions a bit, by describing the suggested solutions more carefully and, if possible, suggesting additional ones (e.g., clear guidelines on how to avoid data leakage).

We agree that adding a more detailed discussion of possible solutions will increase the quality of this manuscript. As such, we have expanded our discussion as follows (page 16, line 8):

“Numerous strategies can help prevent leakage in neuroimaging and other machine-learning applications. These strategies include carefully developing and sharing code, alternative validation strategies, model information sheets (Kapoor and Narayanan, 2023), skepticism about one’s own results, and cross-disciplinary collaborations. Writing and maintaining code should incorporate several facets to reduce the likelihood of leakage, including establishing an analysis plan prior to writing code, using well-maintained packages, and sharing code. One’s analysis plan should be set ahead of time, either informally or, if appropriate, formally through pre-registration. As one tries more pipelines, especially if searching for a significant result (i.e., “p-hacking”), leakage is more likely to occur. A predefined plan could minimize the likelihood of leakage by detailing how features will be selected, which models will be trained, and how possible covariates and nested structures will be handled. Another suggestion for reducing the likelihood of leakage is using well-maintained packages. For example, Scikit-learn has a k-fold cross-validation package (Pedregosa *et al.*, 2011) that has been thoroughly tested, whereas developing k-fold cross-validation code from scratch may lead to accidental leakage. Among many other benefits, sharing code, particularly well-documented code, could decrease the effects of leakage by allowing external reviewers to investigate published pipelines for leakage. Relatedly, although not always possible, distributing preprocessed data can make the reproduction of results much easier and less time-consuming for reviewers or those who want to verify the validity of a predictive model.

Moreover, most neuroimaging papers are evaluated with train/test splits or k-fold cross-validation. However, alternative validation strategies, such as a lockbox (Hosseini *et al.*, 2020) or external validation, may reduce the likelihood of leakage. Both these strategies help to maintain a clearer separation between training and test data, where a lock box entails leaving out a subset of the data until a final evaluation (Hosseini *et al.*, 2020) and external validation consists of applying a model to a different dataset. Another strategy to decrease the prevalence of leakage is using model information sheets, such as the one proposed by Kapoor and Narayanan (Kapoor and Narayanan, 2023). Model information sheets allow for the authors, reviewers, and public to reflect upon the work and identify possible leakage. However, it may be difficult to verify the accuracy of model information sheets when data cannot be shared (Kapoor and Narayanan, 2023). This limitation is especially true for neuroimaging datasets, which often require applications to access the data. As a result, we also recommend healthy skepticism of one’s results. For instance, if a machine learning pipeline leads to a surprising result, the code should be scrutinized by asking a collaborator to view one’s code or repeat the analyses on synthetic data. Finally, collaborations across disciplines to incorporate domain and machine learning experts will help prevent leakage (Kapoor and Narayanan, 2023). Domain experts can bring knowledge of the nuances of datasets (e.g., the prevalence of family structures in neuroimaging datasets). In contrast, machine learning experts can help domain experts train models to avoid leakage.”

Minor:

1. *The length of the resting-state data was different across datasets (26 minutes for HCPD, 10 minutes for ABCD and HBN, and 6 minutes for PNC). Could this difference affect the results across the four datasets?*

The length of the scans could drive differences in the prediction performance across datasets, but we do not expect that it would affect the leakage analyses we ran in this study. We added the following sentence to the limitations section (page 17, line 10):

“Also, differences in scan lengths between datasets may drive differences in performance across datasets. However, it should not affect the main conclusions of this paper regarding leakage in machine learning models.”

2. Line 14 page 13: “Excluding covariate regression (median $r_{coef}=0.31-0.86$), site correction (median $r_{coef}=0.86-0.99$), or both (median $r_{coef}=0.32-0.82$) resulted in moderate changes in the coefficients.” This sentence sounds like either excluding covariate regression or excluding site correction had similar effects on the model coefficients, while in fact the effect of excluding covariate regression was much stronger (the ranges of medians do not even overlap). This should be more clearly stated.

We have updated this sentence as follows (page 8, line 17):

“Excluding site correction (median $r_{coef}=0.75-0.99$) led to minor coefficient changes. Meanwhile, excluding covariate regression (median $r_{coef}=0.31-0.84$) or excluding both covariate regression and site correction (median $r_{coef}=0.32-0.81$) resulted in moderate coefficient changes.”

3. Page 3 line 37: I believe there’s a typo, and $N = 4059$ should be $N = 4079$.

Thank you for catching this typo. As mentioned earlier in this response, we have now included an expanded ABCD sample, so the sample sizes have now been changed.

4. Figure 2 caption: “Rows represent different leakage types” this is an error since this is a figure of non-leaky analysis choices, and rows here figure represent different analysis choices.

We have fixed the caption as follows:

“Rows represent different non-leaky analysis choices, and columns show different phenotypes.”

Reviewer #3 (Remarks to the Author):

Many publications that use machine learning report an inflated prediction performance. One of the causes is an incorrect validation scheme, in which the training and testing data are not independent and thus the test data is not really unseen -- information "leaks" into the model training.

In this study, the authors conduct extensive experiments to measure the impact of such incorrect validation schemes on datasets and prediction tasks that have been widely used in the neuroimaging literature. Several typical mistakes in the validation strategy are applied on combinations of 3 prediction targets and 4 datasets subsampled at 4 different sample sizes.

The authors explore 3 classic families of mistakes: fitting part of the pipeline on the full dataset (2.8), errors related to covariate regression (2.9), failing to account for dependencies between data points (2.10).

Some other common errors such as dimensionality reduction on the whole dataset or hyperparameter overfitting are not considered.

The worse cases of prediction performance over-estimation occur for fitting feature selection on the whole data ("feature leakage") and overlapping train and test sets ("subject leakage"). Those results are expected but it is useful to observe them on real neuroimaging data.

The paper is well written, the objectives and experiments are clear and convincing.

Thank you for your review and your recommendations to improve the paper. We incorporated your recommendations in a point-by-point response below.

I would have liked to see more discussion on the following points:

1. The authors correctly write that including in the train set data points that are not independent from testing data points increases scores, and therefore we should be careful to define groups (for example families) that must not be spread between train and test. However the definition of such groups can depend on the particular task at hand. For example, if we want to evaluate a model that should generalize across sites, we should leave out whole sites in the cross validation. However, if the application we envision makes some data from the site where the model will be deployed available during training, not leaving out whole sites may be acceptable. Could the authors comment on what kind of dependencies can exist between data points (shared family, site, subject, ...) and on when we should take care to leave out whole groups?

Thank you for this suggestion. We have added the following paragraph before the conclusion (page 17, line 19):

“Another limitation is that leakage is not always as well-defined as in this paper. Some examples are universally leakage, such as ignoring family structure, accidentally duplicating data, and selecting features in the combined training and test data. In other cases, whether training and test data are independent may depend on the goal. For example, one may wish to develop a model that will be applied to data from a new site, in which case leave-one-site-out prediction would be necessary. Here, leakage would be present if data from the test site were included when training the model. However, other applications, such as those presented in this paper,

may not require data to be separated by site and can instead apply site correction methods. Similarly, if one wishes to demonstrate that a model generalizes across diagnostic groups, the model should be built on one group and tested on another. The application-dependent nature of leakage highlights the importance of attention to detail and thoughtful experimentation in avoiding leakage.”

2. Could the authors discuss a bit more the kinds of leakage they chose to leave out of this study, and how the considered mistakes were chosen: are they more common, more likely to cause inflated scores, etc. I do think the choice made in this study is appropriate because kinds of leakage that involve the target variable (as considered here), rather than only features (eg unsupervised dimensionality reduction on the whole dataset), are far worse.

Thank you for this suggestion. We have added a section describing our selection process to the methods section (page 20, line 18).

4.7 Selection of forms of leakage

Due to the myriad forms of leakage, investigating every type of leakage is not feasible. In this work, we focused on three broad categories of leakage, including feature selection leakage, covariate-related leakage (leakage via site correction or covariate regression), and subject-level leakage (leakage via family structure or repeated subjects). We chose these particular forms of leakage because we expect that they are the most common and/or impactful errors in neuroimaging prediction studies. In our experience, feature selection leakage is an important consideration because it may manifest in subtle ways. For example, one may perform an explanatory analysis, such as determining the most significantly different brain networks between two groups, and then use these predetermined networks as predictive features, which constitutes leakage. As for covariate-related leakage, we have noticed that site correction and covariate regression are often performed on the combined training and test data in neuroimaging studies. Finally, for subject-level leakage, family structure is often ignored in neuroimaging datasets, unless it is being explicitly studied. Thus, understanding how prediction performance may be altered by these forms of leakage remains an important question.

Some forms of leakage were not considered in this work, including temporal leakage, the selection of model hyperparameters in the combined training/test data, unsupervised dimensionality reduction in the combined training/test data, standardization of the phenotype, and illegitimate features. Temporal leakage, where a model makes a prediction about the future but uses test data from a time point prior to the training data (Kapoor and Narayanan, 2023), is not relevant for cross-sectional studies using static functional and structural connectivity, but it could be relevant for prediction studies using longitudinal data or brain dynamics. While evaluating models in the test dataset to select the best model hyperparameters is a form of leakage, it has been previously studied in neuroimaging (Hosseini *et al.*, 2020) and therefore was not included in this study. Various forms of unsupervised dimensionality reduction, such as independent component analysis (McKeown *et al.*, 1998; Chen *et al.*, 2008), are popular in neuroimaging and constitute leakage if performed in the combined training and test datasets prior to performing prediction. Leakage via unsupervised dimensionality reduction should be

avoided, but we felt that leakage via feature selection is a dimensionality reduction technique that is both more common and more impactful, due to its involvement of the target variable. Moreover, phenotypes are sometimes standardized (i.e., z-scored) outside of the cross-validation folds, but this form of leakage was not investigated in this work because it is insensitive to the most common evaluation metrics in prediction using neuroimaging connectivity (Pearson's r , r^2). Finally, leakage via illegitimate features entails the model having access to features which it should not, such as a predictor that is a proxy for the outcome variable (Kapoor and Narayanan, 2023). Studies using both imaging and clinical or phenotypic measures to predict other outcomes should be cautious of leakage via illegitimate features, but it is less relevant for studies predicting strictly from structural or functional connectivity data.

3. Regarding site and covariate regression: I would argue that a better approach would be to not regress covariates out, but compare to baseline models that have only access to the covariates. I am not sure covariate regression is well-founded in a predictive setting or that we can talk about "leakage via illegitimate features" -- in all situations where a machine model would be used it would be possible to have subjects' sex and age. But if we want to check that brain imaging data brings additional information, we can compare to baselines that use only the covariates. For generalization across sites, rather than removing site effects we should evaluate performance on left-out sites. A more detailed discussion can be found in

Dockès, J., Varoquaux, G. and Poline, J.B., 2021. Preventing dataset shift from breaking machine-learning biomarkers. GigaScience, 10(9), p.giab055.

Thank you for raising this interesting point. We agree that covariate regression is not well-founded when the goal is to maximize prediction performance. We have added this idea, as well as removed the notion of neuroimaging covariates being considered illegitimate features. However, we would also like to note that the goal of many neuroimaging prediction studies is to use prediction to explain a brain-behavior relationship, rather than maximize prediction for a clinical application (Rosenberg, Casey and Holmes, 2018). In this case, while comparing a neuroimaging model (no covariate regression) to a baseline covariate model will determine whether brain imaging data brings additional information, it may not accurately represent which brain networks/features relate to the phenotype beyond other demographic variables such as sex and age. We have added these ideas to the limitations section:

Before: "In this work, we considered omission of covariate regression and site correction to be a "non-leaky analysis choice." However, it could arguably be considered leakage via illegitimate features, meaning that the model predicts using features to which it should not have access (Kapoor and Narayanan, 2022)."

After (page 17, line 5): "Alternative methods may be more appropriate for accounting for possible covariates or site differences in a prediction setting, such as comparing neuroimaging data models to models built only using covariates or leave-one-site-out prediction (Dockès, Varoquaux and Poline, 2021). Nevertheless, we still included covariate regression and site correction in our analysis because they are common in the field and may still be well-suited for using prediction to explain the generalizability of brain-behavior relationships."

4. Regarding the recommendations to avoid leakage in future studies: studies that are prone to leakage are likely the same studies that are prone to having code that is complex, poorly written and difficult to understand, and unlikely to be reused. Therefore, I wonder how likely cross-

validation errors in this code would be discovered by reviewers or others, even if the code was made public. Could the authors provide a few more suggestions, perhaps some sanity checks with synthetic data, the use of well-maintained packages for implementing the cross-validation, etc.?

Thank you for raising this point. We have updated the discussion section with several more possible solutions as follows (page 16, line 8):

“Numerous strategies can help prevent leakage in neuroimaging and other machine-learning applications. These strategies include carefully developing and sharing code, alternative validation strategies, model information sheets (Kapoor and Narayanan, 2023), skepticism about one’s own results, and cross-disciplinary collaborations. Writing and maintaining code should incorporate several facets to reduce the likelihood of leakage, including establishing an analysis plan prior to writing code, using well-maintained packages, and sharing code. One’s analysis plan should be set ahead of time, either informally or, if appropriate, formally through pre-registration. As one tries more pipelines, especially if searching for a significant result (i.e., “p-hacking”), leakage is more likely to occur. A predefined plan could minimize the likelihood of leakage by detailing how features will be selected, which models will be trained, and how possible covariates and nested structures will be handled. Another suggestion for reducing the likelihood of leakage is using well-maintained packages. For example, Scikit-learn has a k-fold cross-validation package (Pedregosa *et al.*, 2011) that has been thoroughly tested, whereas developing k-fold cross-validation code from scratch may lead to accidental leakage. Among many other benefits, sharing code, particularly well-documented code, could decrease the effects of leakage by allowing external reviewers to investigate published pipelines for leakage. Relatedly, although not always possible, distributing preprocessed data can make the reproduction of results much easier and less time-consuming for reviewers or those who want to verify the validity of a predictive model.

Moreover, most neuroimaging papers are evaluated with train/test splits or k-fold cross-validation. However, alternative validation strategies, such as a lockbox (Hosseini *et al.*, 2020) or external validation, may reduce the likelihood of leakage. Both these strategies help to maintain a clearer separation between training and test data, where a lock box entails leaving out a subset of the data until a final evaluation (Hosseini *et al.*, 2020) and external validation consists of applying a model to a different dataset. Another strategy to decrease the prevalence of leakage is using model information sheets, such as the one proposed by Kapoor and Narayanan (Kapoor and Narayanan, 2023). Model information sheets allow for the authors, reviewers, and public to reflect upon the work and identify possible leakage. However, it may be difficult to verify the accuracy of model information sheets when data cannot be shared (Kapoor and Narayanan, 2023). This limitation is especially true for neuroimaging datasets, which often require applications to access the data. As a result, we also recommend healthy skepticism of one’s results. For instance, if a machine learning pipeline leads to a surprising result, the code should be scrutinized by asking a collaborator to view one’s code or repeat the analyses on synthetic data. Finally, collaborations across disciplines to incorporate domain and machine learning experts will help prevent leakage (Kapoor and Narayanan, 2023). Domain experts can bring knowledge of the nuances of datasets (e.g., the prevalence of family structures in neuroimaging datasets). In contrast, machine learning experts can help domain experts train models to avoid leakage.”

References

- Achenbach, T.M. and Ruffle, T.M. (2000) 'The Child Behavior Checklist and related forms for assessing behavioral/emotional problems and competencies', *Pediatrics in review / American Academy of Pediatrics*, 21(8), pp. 265–271. Available at: <https://doi.org/10.1542/pir.21-8-265>.
- Chen, S. *et al.* (2008) 'Group independent component analysis reveals consistent resting-state networks across multiple sessions', *Brain research*, 1239, pp. 141–151. Available at: <https://doi.org/10.1016/j.brainres.2008.08.028>.
- Dockès, J., Varoquaux, G. and Poline, J.-B. (2021) 'Preventing dataset shift from breaking machine-learning biomarkers', *GigaScience*, 10(9). Available at: <https://doi.org/10.1093/gigascience/giab055>.
- Hosseini, M. *et al.* (2020) 'I tried a bunch of things: The dangers of unexpected overfitting in classification of brain data', *Neuroscience and biobehavioral reviews*, 119, pp. 456–467. Available at: <https://doi.org/10.1016/j.neubiorev.2020.09.036>.
- Kapoor, S. and Narayanan, A. (2022) 'Leakage and the Reproducibility Crisis in ML-based Science', *arXiv [cs.LG]*. Available at: <http://arxiv.org/abs/2207.07048>.
- Kapoor, S. and Narayanan, A. (2023) 'Leakage and the reproducibility crisis in machine-learning-based science', *Patterns (New York, N.Y.)*, 4(9), p. 100804. Available at: <https://doi.org/10.1016/j.patter.2023.100804>.
- McKeown, M.J. *et al.* (1998) 'Analysis of fMRI data by blind separation into independent spatial components', *Human brain mapping*, 6(3), pp. 160–188. Available at: [https://doi.org/10.1002/\(SICI\)1097-0193\(1998\)6:3<160::AID-HBM5>3.0.CO;2-1](https://doi.org/10.1002/(SICI)1097-0193(1998)6:3<160::AID-HBM5>3.0.CO;2-1).
- Pedregosa, F. *et al.* (2011) 'Scikit-learn: Machine learning in Python', *The Journal of machine Learning research*, 12, pp. 2825–2830. Available at: https://www.jmlr.org/papers/volume12/pedregosa11a/pedregosa11a.pdf?source=post_page-----
- Rosenberg, M.D., Casey, B.J. and Holmes, A.J. (2018) 'Prediction complements explanation in understanding the developing brain', *Nature communications*, 9(1), p. 589. Available at: <https://doi.org/10.1038/s41467-018-02887-9>.
- Scheinost, D. *et al.* (2019) 'Ten simple rules for predictive modeling of individual differences in neuroimaging', *NeuroImage*, 193, pp. 35–45. Available at: <https://doi.org/10.1016/j.neuroimage.2019.02.057>.
- Yeung, A.W.K. *et al.* (2022) 'Reporting details of neuroimaging studies on individual traits prediction: A literature survey', *NeuroImage*, 256, p. 119275. Available at: <https://doi.org/10.1016/j.neuroimage.2022.119275>.

Reviewer #1 (Remarks to the Author):

The paper describes the effects of data leakage in predictive models in the field of neuroimaging. The authors consider multiple datasets, multiple forms of leakage, and multiple model types. In general, they find that some forms of leakage inflate predictive performance, sometimes drastically. The findings are a valuable reminder of the importance of avoiding leakage. They also help identify types of leakage that are particularly harmful. Leakage is never a correct practice, as the authors emphasize. Still, given that leakage in published work is already widespread, it will be important to know which previously reported results are likely to be substantially impacted, and these findings help toward that goal.

My previous review included several suggestions and questions, all of which have been satisfactorily addressed in the revision.

Thank you for the updates to the paper and the helpful summary of changes.

Reviewer #1 (Remarks on code availability):

I was not able to run the code, because it contained references to .mat files that didn't seem to be available in the repository.

Reviewer #2 (Remarks to the Author):

I thank the authors for fully addressing my comments. I have no further comments.

REVIEWER COMMENTS

We thank all the reviewers for their thoughtful comments and help in improving this manuscript.

Reviewer #1 (Remarks to the Author):

The paper describes the effects of data leakage in predictive models in the field of neuroimaging. The authors consider multiple datasets, multiple forms of leakage, and multiple model types. In general, they find that some forms of leakage inflate predictive performance, sometimes drastically. The findings are a valuable reminder of the importance of avoiding leakage. They also help identify types of leakage that are particularly harmful. Leakage is never a correct practice, as the authors emphasize. Still, given that leakage in published work is already widespread, it will be important to know which previously reported results are likely to be substantially impacted, and these findings help toward that goal.

My previous review included several suggestions and questions, all of which have been satisfactorily addressed in the revision.

Thank you for the updates to the paper and the helpful summary of changes.

Thank you for your comments and time in reviewing our manuscript.

Reviewer #1 (Remarks on code availability):

I was not able to run the code, because it contained references to .mat files that didn't seem to be available in the repository.

Unfortunately, we are unable to share the .mat files, which contain connectome data, due to data sharing restrictions. We clarified the code availability statement as follows:

“Please note that the connectome files cannot be shared due to data sharing restrictions, and thus we do not provide data for running the main analysis code.”

Reviewer #2 (Remarks to the Author):

I thank the authors for fully addressing my comments. I have no further comments.

Thank you for your comments and time in reviewing our manuscript.